# Moving event detection from LiDAR point streams

Huajie Wu [1,2], Yihang Li[1,2], Wei Xu[1], Fanze Kong[1] & Fu Zhang[1]✉

In dynamic environments, robots require instantaneous detection of moving events with microseconds of latency. This task, known as moving event detection, is typically achieved using event cameras. While light detection and ranging (LiDAR) sensors are essential for robots due to their dense and accurate depth measurements, their use in event detection has not been thoroughly explored. Current approaches involve accumulating LiDAR points into frames and detecting object-level motions, resulting in a latency of tens to hundreds of milliseconds. We present a different approach called M-detector, which determines if a point is moving immediately after its arrival, resulting in a point-by-point detection with a latency of just several microseconds. M-detector is designed based on occlusion principles and can be used in different environments with various types of LiDAR sensors. Our experiments demonstrate the effectiveness of M-detector on various datasets and applications, showcasing its superior accuracy, computational efficiency, detection latency, and generalization ability.

Autonomous robots, including self-driving vehicles and autonomous drones, have the potential to revolutionize various applications such as last-mile delivery, robotaxi, agriculture, and aerial imaging, making them increasingly relevant to our daily lives. However, one of the key challenges in deploying these robots is the detection and avoidance of non-cooperative moving objects that are prevalent in real-world environments. Accidents caused by collisions with fast-moving objects, such as tossed objects or birds that collide with drones[1,2], and sudden-crossing pedestrians or cyclists hit by self-driving vehicles[3–5], are examples of these challenges. To avoid such accidents, a robot must detect fast-moving objects or any of their moving parts immediately after the occurrence of the motion. This task is known as moving event detection, or event detection.

The task of event detection is usually achieved by, and also obtains its name from, event cameras[6]. These cameras are designed to detect changes in a scene with a reaction time of microseconds. Unlike traditional cameras, which measure the intensity of a pixel at a fixed rate (i.e., the frame rate), an event camera measures the change in intensity (rather than the intensity itself) of a pixel. When a change in intensity is detected at a pixel, such as due to moving objects, an event is triggered at that pixel to indicate a change in the scene. This

generates a stream of asynchronous events with microsecond-level latency. Due to the high dynamic range, low power consumption, and low detection latency[7], event cameras have been used in many interesting applications such as dynamic obstacle avoidance for quadrotors[8], video reconstruction in high-speed motion[9], and visual-inertial odometry for extreme motion conditions[10].

Light detection and ranging (LiDAR) sensors are another type of sensors widely used for autonomous robots. Unlike cameras that measure the intensity or intensity change at each pixel, LiDAR sensors measure the depth of that pixel by emitting a laser pulse along the pixel direction and computing the laser time of flight (ToF). Such an active and direct ranging mechanism can produce depth measurements that are very accurate (e.g., centimeter level), efficient (e.g., without extra triangulation for depth estimation), and independent of external illumination, enabling the robot to perceive its surrounding environments precisely and timely even at night. Moreover, LiDAR sensors often have tens to hundreds of laser emitters stacked in an array and each one emits laser pulses at microseconds interval[11,12], producing tens of thousands to millions of points per second. These high-frequency point measurements, although at a fixed rate, has a temporal resolution of microseconds to sub-microseconds that is similar to event cameras.

[1]Department of Mechanical Engineering, The University of Hong Kong, Pokfulam, Hong Kong 999077, China. [2]These authors contributed equally: Huajie Wu, Yihang Li. ✉e-mail: fuzhang@hku.hk

Fully exploiting these high-frequency measurements could provide extremely timely detection of any moving events in the scene. Specifically, it requires a moving point to be detected immediately after its arrival to minimize the latency. This online detection of moving events at the rate of point sampling is analogous to event cameras and hence referred to as event detection.

Moving event detection could be achieved at the measuring stage of a LiDAR sensor, such as the Frequency-Modulated Continuous Wave Laser Detection and Ranging (FMCW-LADAR) sensors. Compared to the standard ToF LiDAR, FMCW-LADAR involves a continuously emitted laser beam and utilizes the Doppler effect to acquire information on range and velocity[13]. While being able to measure the velocity of each measured point, FMCW-LADAR can only measure the point velocity component along a laser ray, failing to detect any movements perpendicular to the ray[14]. Besides, the FMCW-LADAR scene acquisition time could take several times longer than LiDAR's[15], which restricts its frame rate that is crucial for robotics applications. Furthermore, the current FMCW system requires a fairly large processing unit, leading to a system of bigger size, weight, and power (SWaP) requirements[13].

Standard ToF LiDARs have gained much wider applications in robotics due to their rapidly decreasing cost and requirements for SWaP. Nevertheless, there are few works addressing the task of event detection using ToF LiDAR measurements. A relevant task that has caught much research attention is moving object segmentation (MOS), which aims to segment points on moving objects from a LiDAR frame. Existing works on MOS are mainly based on consistency check[16–19], occupancy map[20–23], semantic segmentation[24–29], and motion segmentation[30–35]. Consistency check methods[16–19] typically compare points in a new frame to points from previous frames, labeling inconsistent measurements as on moving objects. Occupancy map methods[20–23] often build an occupancy-grid map and label points in a known-free grid as moving objects. Semantic segmentation methods[24–29] first segment the LiDAR points, and then determine the points' moving status based on their segmented class. For example, points belonging to vehicles or pedestrians are labeled as moving and points belonging to walls or traffic lights are labeled as static. This labeling mechanism can detect movable objects (e.g., a parked car) rather than truly moving ones as required by the MOS task. To address this issue, recent works focus on training neural network classifiers[30–35] to segment true moving objects in a frame by enforcing the neural networks to learn the movements instead of appearance contained in the input LiDAR points.

Existing works on moving object segmentation cannot adequately fulfill the task of event detection for a few reasons. First, existing methods operate on frames at a rather low frequency (e.g., 10 Hz), requiring the accumulation of LiDAR points into frames. This accumulation causes an obvious delay that is equal to the frame interval, typically 100 ms, which nullifies the inherent high-rate sampling feature of LiDAR sensors. In addition, ref. 19 relies on a future frame for detecting moving objects, which exacerbates the delay further. For methods that employ occupancy grid maps[20–23], constructing the occupancy map is often computationally and memory intensive, particularly when detecting movements over long ranges (e.g., hundreds of meters) or with small movements (e.g., a few centimeters), making real-time operation challenging. Moreover, ref. 21–23 are offline systems that require all LiDAR frames for occupancy map construction. For learning methods[26–35], they often require a substantial amount of ground-truth-labeled data for training neural networks, which can be difficult to obtain, especially given the diverse range of LiDAR types used in the robotics community. Generalization to classes, LiDAR types, and scenes not included in the training data remains a significant challenge.

Our method for moving event detection makes full use of the high-rate sampling of a ToF LiDAR sensor by detecting the movement of each point immediately upon its arrival. This point-by-point detection mechanism leads to an online system operating on the LiDAR point streams. Moreover, it eliminates the need for frame accumulation and

achieves a detection latency of just 2–4 $\mu$s per point. Unlike existing methods that rely on the appearance of objects in the input point cloud[26–35], our method uses the motion cues of each point. This provides two key benefits: first, it detects moving, rather than movable, objects (Fig. 1b); second, it detects any moving object, or any moving part of the object, regardless of its shape or class (Fig. 1a). As a model-based method that explicitly exploits the motion cues, our method can be easily adapted to different object classes, LiDAR types, and scenes.

We term our method as M-detector (Supplementary Video 6), which stands for moving event detector. The low detection latency of the M-detector also resembles the Magnocellular cell (M-cell) located in the lateral geniculate nucleus (LGN) of human visual systems, which is a specialized "motion-sensitive" cell that detects changes in the visual field with rapid response time due to their larger sizes and faster processing capabilities compared to other cells in the LGN[36] (Fig. 2).

The motion cue that enables M-detector is the occlusion principle (Fig. 3), where an object that crosses the laser rays of a LiDAR sensor will occlude the background that was visible in the past (Fig. 3a) and an object that moves along the laser rays will recursively occlude (or be occluded by) itself (Fig. 3b, c). The occlusion principles hold as long as the sensor returns points collected on the first visible object, which is true for existing robotic LiDARs. Building on these basic physical principles, the M-detector is highly generalizable without requiring massive ground-truth labels for training.

The system design of M-detector is depicted in Fig. 4. The system accepts either an individual point or a frame of points as its input, with the sensor ego-motion compensated in advance. In the case of a frame input, the system serializes the frame into a stream of individual points. Each point, whether input directly or serialized from a frame, undergoes three parallel tests (i.e., event detection) corresponding to the three occlusion principles (Fig. 3). The point is labeled as a moving event if any of the three tests are positive or else labeled as a non-event point (see Methods). The labeled point is then output immediately, leading to a detection delay equal to the processing time of the event detection. Then, the currently labeled points are accumulated for a certain time period, where clustering and region growth are performed to improve the detection results by rejecting possible outliers and accepting further inliers (see Methods). The accumulated points with event labels are finally transformed into a depth image, which is saved in the depth image library for use in detecting future points. The system can also output the event labels as frames after clustering and region growth are completed. This mode of output increases the accuracy but introduces a longer delay due to the time required for point accumulation, clustering and region growth. A more detailed explanation for M-detector's workflow can be found in Supplementary Notes 2.

We demonstrated the effectiveness of the proposed M-detector in terms of detection accuracy, computation efficiency, success rate (Supplementary Notes 3), and practical applications. Results on multiple open datasets, including *KITTI*[37], *SemanticKITTI*[38], *Waymo*[39], and *nuScenes*[40], as well as our in-house dataset, *AVIA-Indoor*, showed that the M-detector achieved consistently higher accuracy while consuming the least computation time among the benchmarked methods. Notably, it ran in real-time and achieved a detection latency of just 2–4 $\mu$s for each received LiDAR point. The method also demonstrated a high level of generalizability, successfully detecting moving objects of various sizes, colors, and shapes in different environments (e.g., laboratory, road, park, at night, see Supplementary Notes 4) and with different LiDAR types (e.g., multi-line spinning LiDARs, non-repetitive scanning LiDARs). Moreover, the high detection accuracy, low detection latency, and training data-free nature of M-detector provide timely and robust moving event detection at a low cost (Supplementary Video 1), which could benefit a substantial number of real-world robotic applications. We demonstrated several practical applications of the M-detector, including moving object detection in autonomous

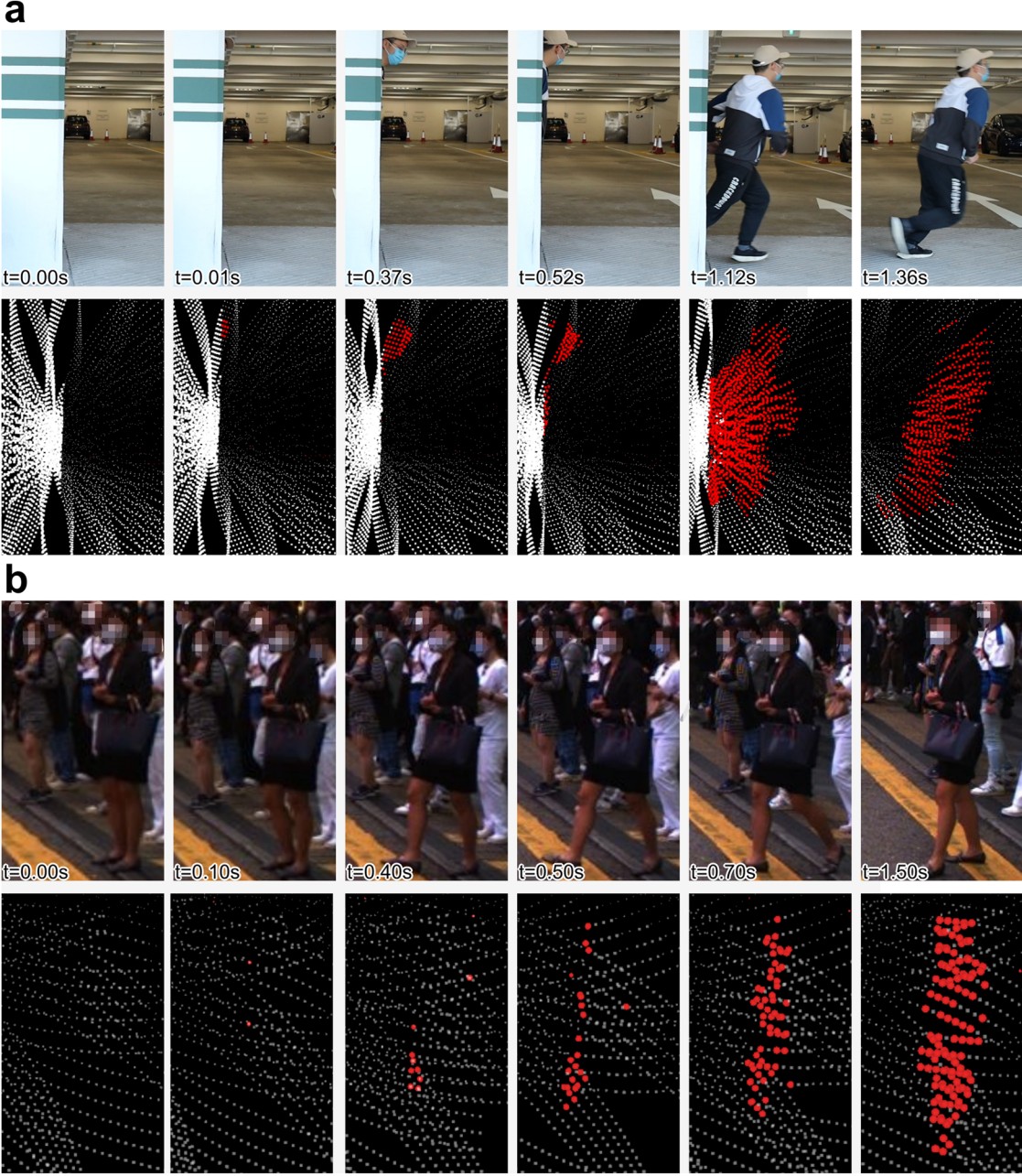

**Fig. 1 | M-detector detects the movements in a scene instantaneously.**
**a** Detection of a sudden-crossing pedestrian. The pedestrian emerged from behind a wall at 0.01 s and quickly entered the road without checking the road condition. **b** Detection of a stationary pedestrian starting to move. The lady waited at an intersection until 0.4 s, when she lifted her leg and began to cross the street. The other pedestrians remain stationary and have not begun moving. In both cases, the upper row shows the sequence of images (for visualization only). The lower row shows the detection results over the last 100 ms of M-detector on a Livox AVIA LiDAR, which has a non-conventional non-repetitive irregular scanning pattern. M-detector determines whether a point is on a moving object right after the point's arrival, leading to a detection latency less than the LiDAR point sampling interval (a few microseconds). Designed on a first principle, the occlusion principle, M-detector detects points sampled on any moving part of the scene regardless of its shape.

driving, avoidance of fast-moving obstacles in quadrotor navigation, vehicle counting in traffic monitoring, intruder detection in surveillance, and dynamic points removal in LiDAR mapping. The M-detector allowed the quadrotor to avoid obstacles with relative speeds up to 7.6 m/s even when the obstacle has a similar color to its background.

## Results

We evaluate the accuracy and computation efficiency of M-detector on three open datasets, including *KITTI*[37], *Waymo*[39], and *nuScenes*[40], and one in-house dataset, *AVIA-Indoor* (Supplementary Notes 5), leading to a total number of 119 sequences and a total duration over 51 min

(Supplementary Table 1). These datasets cover various types of LiDARs (e.g., 32 lines, 64 lines, and non-repetitive irregular scanning), scenes (urban, residential, highway, and indoor laboratory), objects to be detected (vehicles, pedestrians, cyclists, and tossed objects of different sizes or colors (Supplementary Fig. 1)), platforms (e.g., road vehicles, handheld device), and illumination conditions. More details of the datasets and their labeling are shown in Supplementary Notes 5 and Supplementary Notes 6.

In all evaluations, we compared the M-detector with two methods, LMNet[30], which is a representative learning-based motion segmentation method, and SMOS[20], a representative method based on

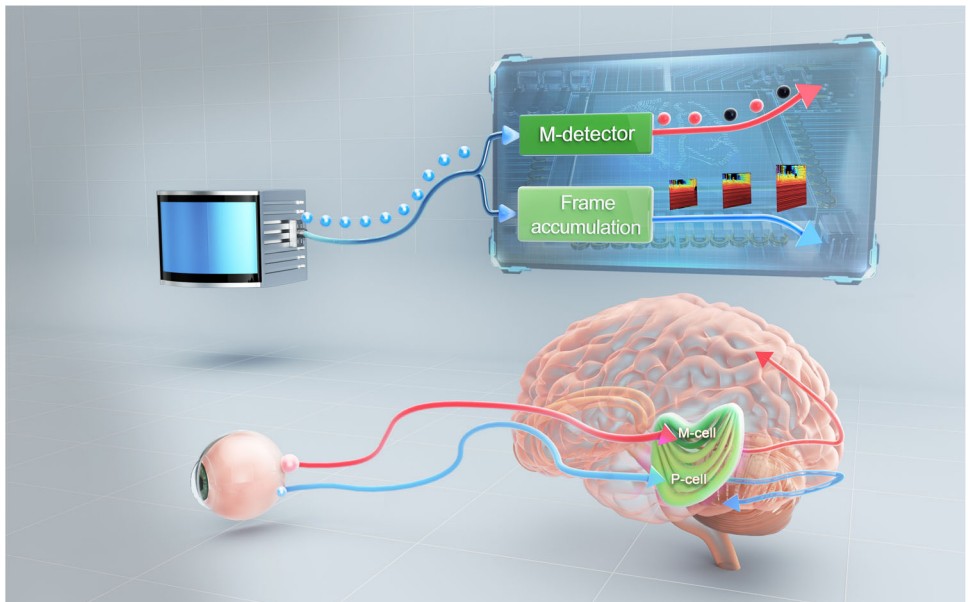

**Fig. 2 | Analogy of M-detector to M-cell.** The M-detector detects moving events from the stream of LiDAR points in a point-by-point manner, leading to a detection latency of just 2-4 $\mu$s. The low latency of M-detector is similar to the Magnocellular cells (M-cells) in the lateral geniculate nucleus (LGN) of human visual systems[61], which also have a fast response time but a low resolution. In contrast, accumulating points into frames leads to a higher resolution but also a longer delay (e.g., 100 ms), a phenomenon similar to the Parvocellular cells (P-cells) in the LGN. This figure is created with the help from a paid third party (https://www.shiyanjia.com).

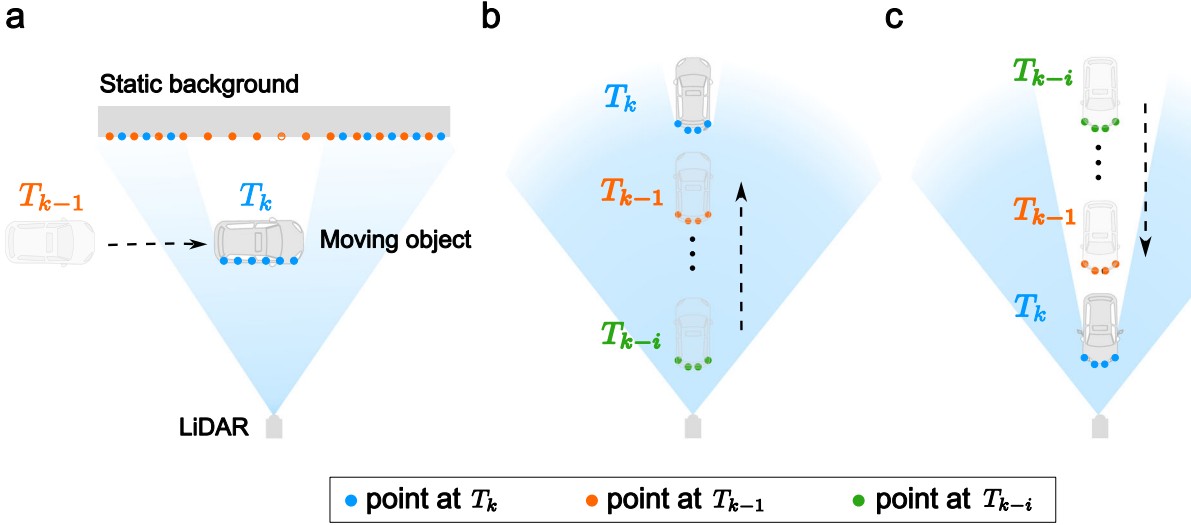

**Fig. 3 | Principle of occlusion. a** When an object is crossing the LiDAR's laser rays, the current points (blue points) will occlude the previous points (orange points) collected at time $T_{k-1}$. **b** When an object is moving along laser rays and away from the sensor, the current points (blue points) will be occluded by all previous points (i.e., orange points at $T_{k-1}$ and green points at $T_{k-i}$) that are further occluded by themselves (i.e., orange points at $T_{k-1}$ are occluded by green points at $T_{k-i}$). **c** When an object is moving along laser rays and towards the sensor, the current points (blue points) will occlude all previous points (i.e., orange points at $T_{k-1}$ and green points at $T_{k-i}$) that further occlude themselves (i.e., orange points at $T_{k-1}$ occlude green points at $T_{k-i}$).

occupancy map. Methods based on semantic segmentation and consistency check are not benchmarked. The reason is that methods based on semantic segmentation only segment movable objects, not moving ones, and thus have a different objective from M-detector. For consistency check-based methods, to the best of our knowledge, there are no open-source implementations available for a fair comparison. We used two variants of LMNet, namely LMNet-1 and LMNet-8*, and the author-tuned parameters of SMOS, as detailed in Supplementary Notes 7. For M-detector, different parameters were adjusted for different LiDAR sensors (Supplementary Table 2) based on the tuning guidelines provided in Supplementary Notes 8. For each type of LiDAR

used in one dataset, the parameters were kept the same for all sequences.

Since all datasets were collected in the frame mode, where points were packed into frames at certain frequencies (Supplementary Table 1), M-detector serialized a frame into a stream of points and labeled the movement of each point sequentially (Fig. 4). In contrast, LMNet and SMOS were both based on frames, where they leveraged all information in the frame to label each point. To make a fair comparison, in addition to the point-out mode of the M-detector, we also compare its frame-out mode, where the event labels were output after the step of clustering and region growth (Fig. 4). The

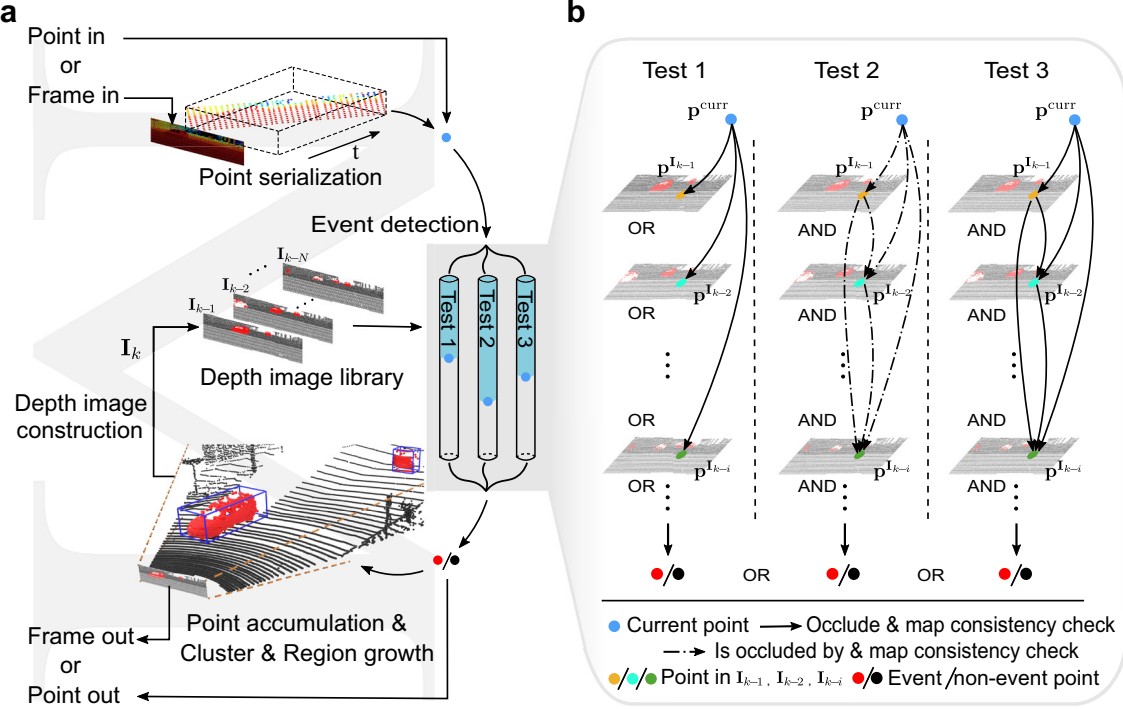

**Fig. 4 | System workflow of M-detector and details of the event detection step. a** The system workflow. **b** The three tests in the event detection.

odometry for ego-motion compensation in all methods is based on FAST-LIO2[41].

## Accuracy

To quantify the accuracy, we adopted a commonly used metric for Moving Object Segmentation (MOS), the Intersection-over-Union (IoU)[42], which is defined below

$$IoU = \frac{TP}{TP + FP + FN} \quad (1)$$

where TP, FP, FN denotes the number of true positive, false positive, and false negative points on moving objects over all sequences in a dataset.

Figure 5a shows the comparison among M-detector, LMNet, and SMOS in terms of the IoU. On the *KITTI* dataset where LMNet and SMOS were both well trained or tuned, they were still outperformed by M-detector (frame-out mode) with large margins. Specifically, M-detector outperformed SMOS by 263.9% (0.746 versus 0.205) and LMNet-8* by 17.5% (0.746 versus 0.635) despite the lack of semantic information used by LMNet-8*. Without such semantic information (i.e., LMNet-1), LMNet's performance dropped even more, achieving an IoU even lower than the point-out mode of M-detector although it used one entire frame to label each point, while the M-detector labeled each point sequentially. The performance degradation of LMNet was mainly due to false detection caused by static pedestrians or cars and missing detection on partially occluded moving cyclists (Fig. 5b). This is reasonable because it was difficult to enforce the neural network classifiers in LMNet to learn the objects' movements rather than appearances. The trained neural networks often falsely detected objects that were movable but not actually moving. In contrast, the M-detector, by design, is sensitive to only objects' movements instead of their appearance, shape, size, etc. As a consequence, the M-detector is able to distinguish an object in moving from that at stationary, regardless of its appearances (e.g., pedestrians, cars, partially occluded cyclists) (Fig. 5b). For SMOS, many missing detection occurred due to its design

(Fig. 5b). The method used the number of points in a voxel to determine the voxel's moving status, which was not effective for objects with movements less than the voxel size or objects at far with fewer measured points (they set the detection range to 25 m in *KITTI*). These two factors deteriorated the performance of SMOS considerably.

For the rest three datasets, the performance of LMNet dropped drastically because its training data do not contain the LiDAR types (e.g., Velodyne HDL-32E, Waymo-64, and Livox AVIA) or objects (e.g., tossed balls) of these datasets. The model-based method, SMOS, performed better than LMNet on the three datasets with our fine parameter tuning, but the overall performance was still quite low. In contrast to LMNet and SMOS, M-detector, both point-out mode and frame-out mode, achieved consistently high IoU on these datasets and outperformed LMNet (LMNet-1 and LMNet-8*) and SMOS with large margins, demonstrating that the M-detector can maintain high accuracy across different types of LiDAR (e.g., multi-line spinning (Fig. 5b) and incommensurable scanning (Fig. 5c)) and scenes (urban, residential, highway and indoor laboratory). It is also noted that the M-detector has an IoU that is higher on *nuScenes* than on the other datasets. This is because *nuScenes* has many dynamic points collected on the ego-vehicle, which is detected by the M-detector. A large number of event points contributed to a high TP and hence a high IoU according to Eq. (1).

## Time consumption and detection latency

We evaluated the computation efficiency of M-detector in terms of the time consumed by processing each frame (or point), including event detection, clustering and region growth, and depth image construction (Fig. 4a). The evaluation was conducted on a desktop with a central processing unit (CPU) of Intel i7-10700 (2.90 GHz, 8 cores) and 48 GB random access memory (RAM). We further compared the computation time to LMNet on the same computation platform. Since LMNet involves neural networks which are best performed on Graphics Processing Unit (GPU), we also ran it on a desktop with an Intel Xeon(R) Gold 6130 CPU (2.10 GHz, 16 cores), a RAM of 64 GB, and a single GeForce RTX 2080 Ti graphics card.

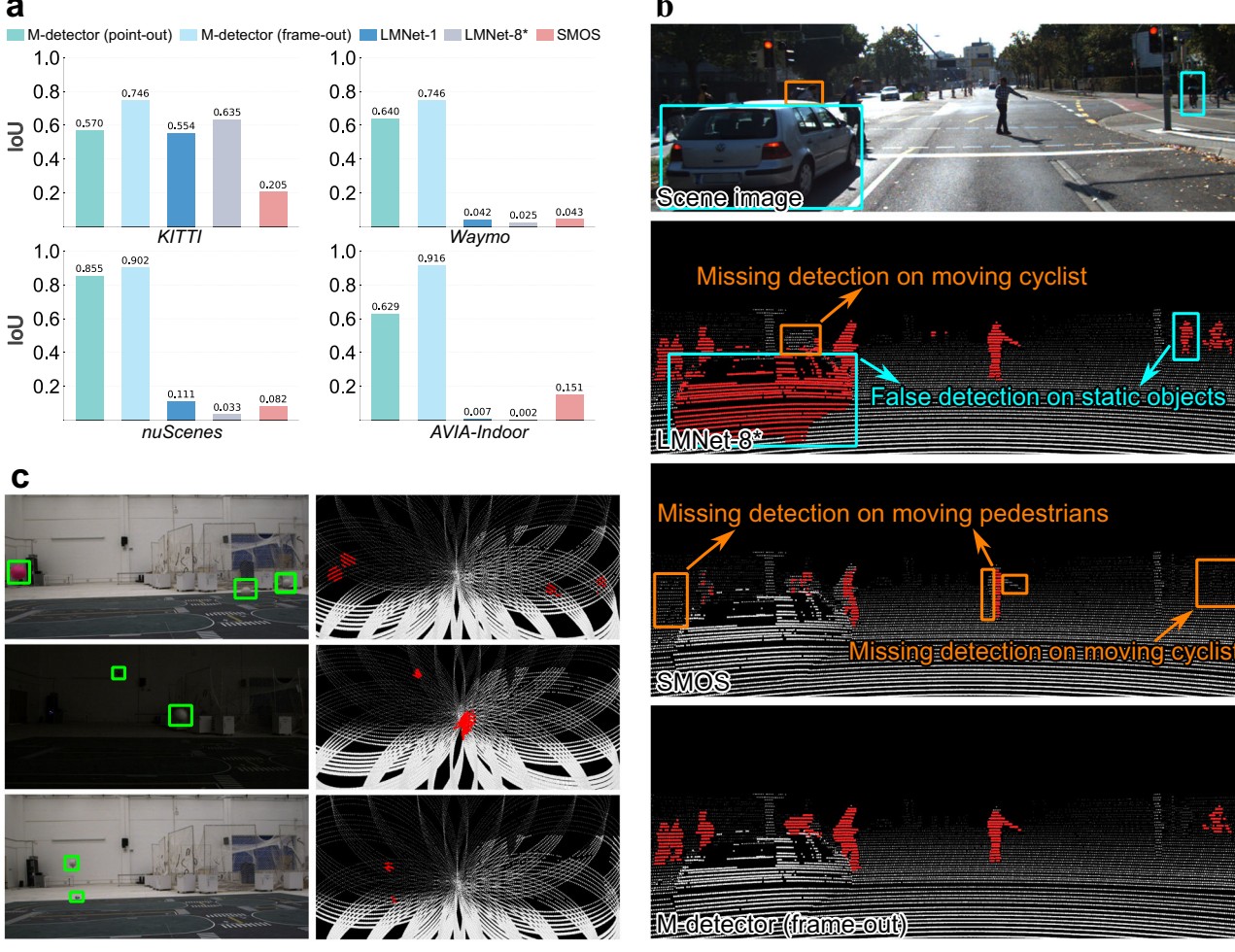

**Fig. 5 | The performance of M-detector on different datasets. a** The IoU results of different methods on different datasets. **b** The comparison among M-detector, LMNet-8*, and SMOS on *KITTI*. The scene image (for visualization only) is the 68th frame in sequence 15. **c** Details of the *AVIA-Indoor* dataset. The left column shows images captured in the scene, and the right column shows the corresponding detection results of M-detector in frame-out mode. In all detection results shown above, the red points represent event points labeled by the method and the white points represent non-event points. Boxes in the images and detection results are manually labeled to highlight the regions of interest.

Figure 6a shows the time consumption of M-detector and LMNet. Overall, M-detector consumed only around 1–6% computation time of LMNet-1 when running on the same CPU, and only around 20–60% of LMNet-1 even when LMNet-1 additionally used a GPU. The resultant time consumption of the M-detector was less than the frame period (87.3 ms versus 100 ms in *KITTI*, 86.2 ms versus 100 ms in *Waymo*, 33.6 ms versus 50 ms in *nuScenes*, and 16.1 ms versus 100 ms in *AVIA-Indoor*), suggesting a real-time running performance. The detailed computation time breakdown of LMNet is shown in the Supplementary Table 3. The inference time of LMNet-1 on GPU was 51.1 ms. Nevertheless, other modules, such as residual image construction, contributed substantial computation time, the total time was 0.8–1.9 times the frame period on the GPU desktop. For LMNet-8*, it infers the points' moving status using eight recent residual images and further leverages the semantic labels generated by SalsaNext. Although the semantic segmentation ran in parallel to the moving events inference, the residual image construction contributed considerably more computation time (Supplementary Table 3). As a result, the mean per-frame computation time of LMNet-8* on CPU was around 10% more than LMNet-1 and on GPU is 200–500% more than LMNet-1, leading to an even larger gap from M-detector. Finally, for the method SMOS, the open-source implementation from[20] was based on MATLAB, whose running time was not optimized for efficiency. As a reference, the computation time of SMOS on the CPU platform was 10-20 times higher than M-detector (615.0 ms in *KITTI*, 2043.9 ms in *Waymo*, 848.7 ms in *nuScenes* and 793.5 ms in *AVIA-Indoor*).

In addition to the computation time, we further evaluated the detection latency, which plays a crucial role in dynamic objects avoidance. The detection latency of M-detector in point-out mode was caused by the event detection and that in frame-out mode was caused by both event detection and clustering & region growth (Fig. 4a). As shown in Fig. 6b, in the frame-out mode, M-detector had a detection latency ranging from 11.5 ms to 65.1 ms due to the varying frame rate and point number in each frame (see further latency breakdown in Supplementary Table 4). In the point-out mode, the detection latency was 2–4 $\mu$s per point, which ensured a low latency detection. In contrast, the detection latency of LMNet and SMOS consisted of the time consumed by all processing steps in their design. Consequently, the values of their detection latency were equal to the computation time, which caused a delay per frame at the level of hundreds of microseconds to seconds.

**Application on autonomous driving**
The detection of sudden crossing pedestrians is a critical challenge in autonomous or assisted driving vehicles, particularly in scenarios where a pedestrian unexpectedly enters the road, providing very little

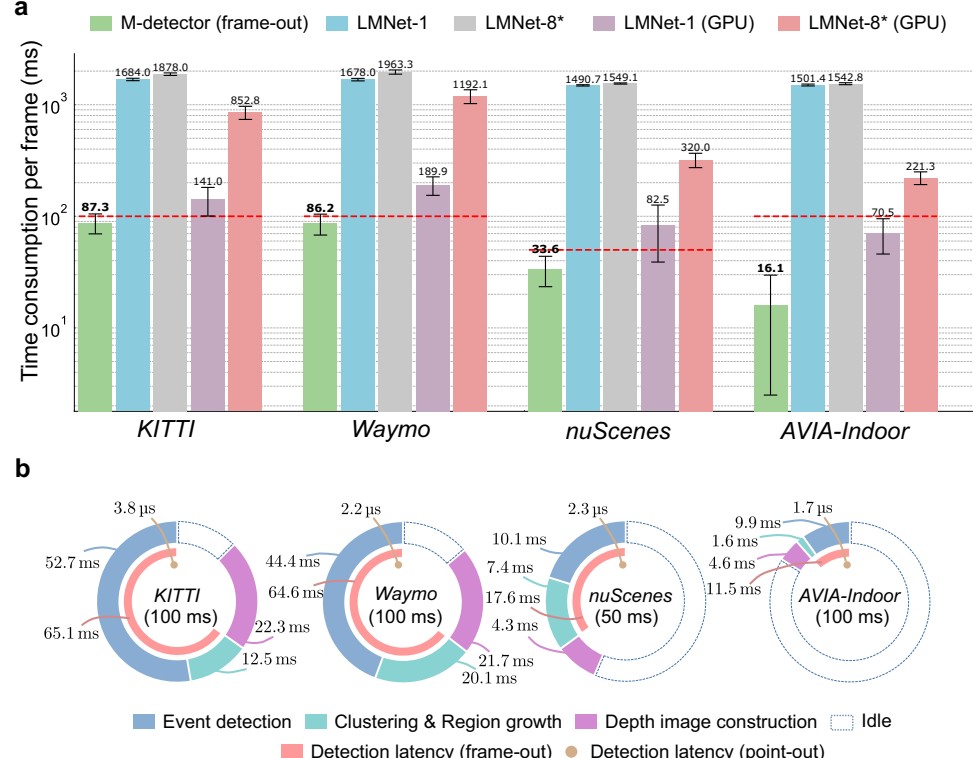

**Fig. 6 | Time consumption and detection latency on different datasets. a** The time consumption per frame of different methods on different datasets. The results for each dataset were obtained by running all frames of all sequences in that dataset. Each bar in the plot denotes the mean time consumption. The two edges around each bar represent one standard deviation among all results in the corresponding dataset and the numbers above the top edges are the value of the mean time consumption. The red dashed lines denote the frame periods of each dataset. Note that the time consumption is shown in log scale. **b** The time consumption breakdown and detection latency of M-detector on different datasets. The outer rings represent the time consumption breakdown, including event detection, clustering & region growth, and depth image construction. The ring's total length denotes the frame period, which is occupied by the above three steps shown in different colors and corresponding numbers. The uncolored portion of each ring denotes the idle time per frame. The middle ring sectors in pink represent the detection latency of the M-detector in frame-out mode, which is caused by both event detection and clustering & region growth. The inner points in brown denote the detection latency of the M-detector in point-out mode, which is caused by event detection of a single point only. The words and numbers at the center of the rings denote the name of the datasets and the corresponding frame period.

time for the vehicle to react. This challenge is further exacerbated when the sudden crossing occurs outside of intersections, as the vehicle may not anticipate such behavior in these areas. According to the National Highway Traffic Safety Administration (NHTSA)[43], 6,516 people in the United States were killed in motor vehicle crashes in 2020, with 75% of these fatalities occurring outside of intersections due to sudden road crossing. The proposed M-detector is particularly suitable for this scenario due to two reasons: (1) it leverages the occlusion principle to detect moving objects of any shape, color, or size; and (2) the method utilizes point-by-point detection (Fig. 4), allowing for microsecond-interval detection of moving objects. These features enable the vehicle to immediately detect crossing pedestrians as soon as any part of the pedestrian enters the sensor's line of sight or begins moving.

Figure 1a shows the case where a pedestrian behind a wall suddenly entered the road. The M-detector successfully detected the pedestrian in 10 ms, among which the major delay was due to the LiDAR scanning (i.e., the time spent on driving the LiDAR prisms to a direction pointing to the moving objects). The processing delay from the receipt of a point to its labeling was only a few microseconds. These point-wise event labels with microseconds-level delay provide extremely timely feedback for a vehicle to make an emergent decision without going through the full perception pipeline that requires more complete measurements (e.g., a complete frame) and a longer processing time, causing a perception delay up to hundreds of milliseconds. Moreover, at the moment the pedestrian was detected, the sensor only measured a very

small part of the pedestrian's head. These partial measurements are very challenging for learning-based methods[26–35] to detect because these methods often leverage the object appearance information and require massive such unusual data to train. Figure 1b shows the case where a lady waiting at an intersection began to cross the street. Initially, she stood still at the intersection. At around 0.4 s, she lifted her leg, which triggered the M-detector to detect two event points immediately. At 0.5 s, she stretched out her leg and triggered more event points. The number of detected event points then increased as her movements became more apparent while no event points were detected on the other pedestrians who have not begun moving. These event labels on moving part of a pedestrian provide timely feedback for the vehicle decision even before the pedestrian has any actual displacement.

### Application on UAV obstacle avoidance

Due to the low detection latency and high computing efficiency, M-detector can be well applied for UAVs with constrained computation resources to evade fast-moving obstacles (e.g., tossed objects, birds). To validate this application, we constructed a quadrotor UAV carrying a Livox AVIA LiDAR and a DJI manifold2-C microcomputer with an Intel i7-8550 U CPU (1.8 GHz, 4 cores) and 8 GB RAM (Fig. 7a). The LiDAR frame rate was set to 50 Hz. Once a LiDAR frame was received, M-detector began detecting the event points. The labeled event points were then sent to the planning module for calculating an evasive trajectory. Details of the system and its modules are presented in Supplementary Notes 9.

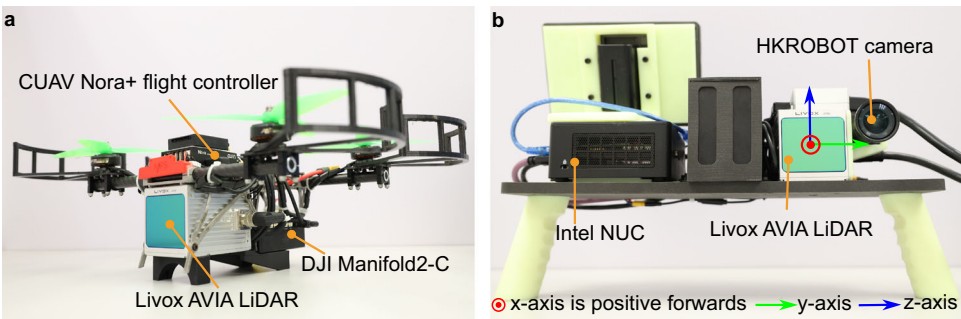

**Fig. 7 | Platforms used in the experiments. a** A small-scale quadrotor equipped with a Livox AVIA LiDAR, a DJI Manifold2-C with an Intel i7-8550 U CPU (1.8 GHz, 4 cores) and 8 GB RAM, and a CUAV Nora+ flight controller. **b** A handheld platform equipped with a Livox AVIA LiDAR, a HKROBOT camera, and an Intel NUC with Intel i7-1260P CPU (2.1 GHz and 12 cores) and 64 G RAM.

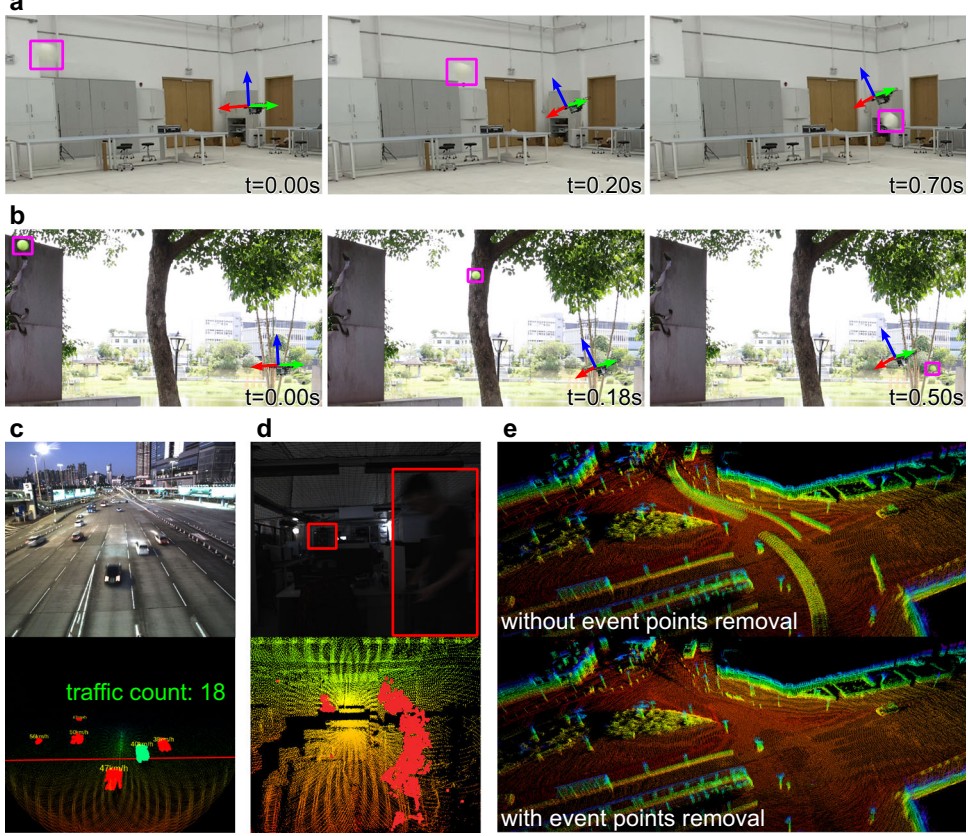

**Fig. 8 | Demonstrations of applications. a** A sequence of the UAV avoiding a thrown ball with a diameter of 20.4 cm in an indoor environment. **b** A sequence of the UAV avoiding a thrown ball with a diameter of 6.5 cm (i.e., a tennis ball) in an outdoor environment. In both (**a**, **b**) to enhance the visualization, a coordinate frame was attached to the UAV and a box was attached to the thrown ball. **c** Vehicle counting at a footbridge. The upper image shows the actual scene and the lower image shows the LiDAR points with detection results. In the lower image, the points colored in red are the detected moving vehicles and the points colored in green indicate the current vehicle being counted with its number shown as "18". **d** Intruder detection at night. Two guests were detected by M-detector (the red points in the lower picture). The red rectangles on the upper picture were added manually for better visualization. **e** Comparison of mapping with and without event points removal on sequence 00 of *KITTI*, accumulated from the 7th frame to the 57th frame. With the M-detector, points due to moving vehicles on the road were successfully detected and removed, leading to a clean point cloud map of the static environments.

Figure 8a shows one of the indoor experiments, where the UAV successfully avoided a thrown ball with a diameter around 20.4 cm. In the experiment, at 0.00 s, the UAV hovered at a stationary position until the detected moving object was within a specified safety threshold (i.e., 4 m). Then from 0.20 s to 0.70 s, the UAV began to avoid the object by moving a certain distance (e.g., 1.5 m) along a direction that is perpendicular to the object incoming direction (Supplementary Notes 9). The speed of the object at the safety threshold was about 7.6 m/s. Figure 8b shows one of the outdoor experiments with a tennis ball of diameter 6.5 cm. The object speed at the safety threshold was about 6.7 m/s. In both experiments, the time for event detection per frame was 1.11 ms on average (Supplementary Table 5), and the total processing time from the receipt of a LiDAR frame to the transmission of an actuator command was around 1.27 ms, comprising 1.11 ms for event detection, 0.13 ms for trajectory planning, and 0.03 ms for control (Supplementary Table 5). The low-latency detection and planning enabled the quadrotor UAV to avoid the thrown obstacles stably. We have also conducted multiple

experiments where the object was tossed when the UAV was in forward flights (Supplementary Video 2). In all the experiments, the UAV avoided the tossed objects and arrived at its prescribed target positions successfully. Additional information of the methods and demonstrations can be found in Supplementary Notes 9.

## Application on traffic monitoring

M-detector could be used for traffic monitoring, which aims to obtain information such as traffic flow and congestion conditions at the macro level and model the behaviors of the road participants (e.g., cars, pedestrians, and cyclists) at the micro level[44,45]. These information is key to traffic planning[46] as well as enhancing the sensing capability of autonomous driving vehicles (e.g., roadside LiDARs[47], V2X techniques[48]). Based on a first principle, the occlusion principle, M-detector can detect moving objects on the road reliably, in real-time, and at a very low cost. To demonstrate this application, we constructed a sensor suite comprising of a Livox AVIA LiDAR and an Intel NUC onboard microcomputer with Intel i7-1260P CPU (2.1 GHz and 12 cores) and 64 G RAM (Fig. 7b) and used it to count the number of passing vehicles on a road. The LiDAR frame rate was set to 10 Hz. Once a LiDAR frame was received, the M-detector began to detect moving points contained in the frame. The labeled points were then clustered, tracked, and counted (Supplementary Notes 10).

Figure 8c showed the results obtained on a footbridge. Once the sensor suite was installed, a straight line on the road below the sensor was calibrated. If a vehicle passed through the line from the prescribed side, the traffic count would increase by one. As shown in Fig. 8c and Supplementary Video 3, the system counted the passing vehicles reliably. The average processing time per frame was 66.99 ms (Supplementary Table 5), which ensured real-time running on the onboard microcomputer.

## Application on surveillance and people counting

M-detector could also benefit applications like surveillance and people counting substantially. For surveillance, due to the active measurements of LiDAR sensors, the M-detector could detect any movements (e.g., caused by intruders) in the scene with low lighting conditions or even at night. One demonstration of the M-detector was shown in Fig. 8d (also see Supplementary Video 4). We used a Livox AVIA LiDAR[49] due to its incommensurate scanning[12], which ensured a more complete scanning within the sensor field of view (FoV). With an onboard microcomputer Intel NUC with Intel i7-1260P CPU (2.1 GHz and 12 cores) and 64 G RAM, the averaging processing time per frame was 47.73 ms, which was shorter than frame period (100 ms at 10 Hz) and suggested a real-time performance. When compared to thermal cameras, LiDAR sensors augmented by the M-detector could serve as a potentially more cost-effective alternative due to emerging low-cost LiDARs. They could also complement thermal cameras by providing direct 3D measurements and detecting non-human moving objects (e.g., drones) that have no thermal difference with surroundings. People counting is an application that aims to improve the space usage or to model human behaviors by analyzing the number of people who visit a place (e.g., workplaces, supermarkets, malls) or the duration of the visit[50,51]. When compared with visual cameras deployed for this application, LiDAR solutions cause fewer privacy concerns from the participants due to the inability to perform facial identification.

## Application on mapping

LiDAR sensors have been widely used for 3D mapping and reconstruction[52–54]. A major problem in LiDAR mapping is the non-existent map points caused by moving objects appearing in the sensor FoV during data collection. M-detector could address this issue by removing points on moving objects in each frame. A clean, moving points-free map could be obtained instantly for preview without any post processing. Fig. 8e shows one demonstration of the M-detector

on *KITTI* sequence 00 (also see Supplementary Video 5). The averaging time of event detection was 84.48 ms (versus the frame period 100 ms at 10 Hz).

## Discussion

Here we discuss the unique features of the M-detector in comparison to existing methods, as well as its potential integration with current robotic techniques. We also discuss the limitations of the M-detector and its implications in robotics applications. Given space constraints, we present only the discussion of the M-detector's new features in the main text. For a more comprehensive discussion, readers are directed to the Supplementary Notes 1.

Among existing moving object segmentation methods, M-detector is unique in both its design principle and exhibited behaviors. Designed on the occlusion principle, M-detector is a model-based method that does not require massive training data of learning-based methods[26–35]. This ensures a high level of generalization to different LiDAR types, object classes, scenes, and carrying platforms without noticeable accuracy degradation. In contrast, the performance of learning-based methods, such as ref. 30, could drop considerably if the test dataset is different from the training dataset. Although re-training the network with samples from the test dataset would improve the accuracy, it would require massive labeled data, which is exactly the bottleneck problem faced by the community. The data labeling problem is even more severe for emerging solid-state LiDARs[12,55] whose scanning patterns are often different from multi-line spinning LiDARs where existing data were labeled.

Besides eliminating the requirement for labeled data, the M-detector is only sensitive to movements in the scene. Consequently, it detects points on objects that are truly moving instead of just movable. In contrast, learning-based methods[26–35] could easily detect points on movable objects. The reason is that a training dataset contains only a few classes of labeled moving objects (e.g., cars, pedestrians, cyclists). Hence it is difficult to enforce the neural networks not to overfit the object appearance but to learn only features of movements. This further causes two problems: one is the false detection of movable but stationary objects, and the other is the missing detection for objects whose appearances were not seen in the training data (e.g., untrained objects, trained objects but partially occluded in the test). These two issues also apply to the semantic segmentation methods for moving objects segmentation[24–29].

When compared to methods based on consistency check[16–19] and occupancy map[20–23], occlusion is a stronger and more efficient clue for movements detection. Consistency check[16–19] cannot distinguish between new points scanned from unseen areas and moving points scanned from seen areas (i.e., the moving points that should be detected), because both cases lead to inconsistencies between current and previous frames. Our occlusion principle does not have such ambiguity since new points scanned from unseen areas would not occlude or be occluded by any points in previous frames and would be always labeled as static (as it should be because it is not possible to decide if an object is moving by looking it only once). Likewise, the occupancy map used in ref. 20–23 can address this ambiguity by distinguishing the unseen areas from seen ones. However, building the occupancy map is rather time-consuming due to the large number of traversed voxels caused by long LiDAR measuring range, large points number, or high map resolution. In contrast, the occlusion principle examines the occlusion relation between current points and map points directly without traversing voxels in between them (see Methods). This leads to a computationally-efficient design that is able to run in real-time on a single CPU. The computation time was only half of learning-based methods (e.g., LMNet[30]) that even leveraged GPU acceleration.

Another unique feature of the M-detector is the ability to detect the eventness of a LiDAR point right after its arrival without

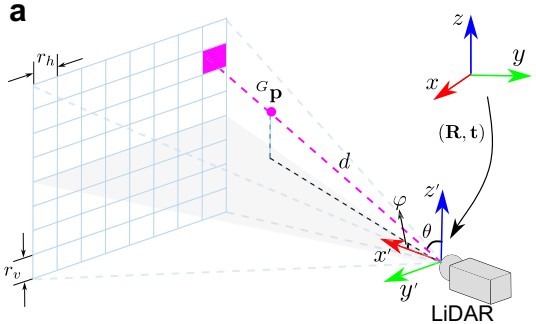

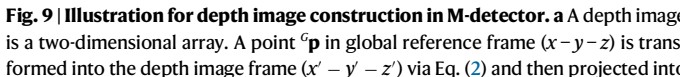

**Fig. 9 | Illustration for depth image construction in M-detector. a** A depth image is a two-dimensional array. A point $^G\mathbf{p}$ in global reference frame $(x-y-z)$ is transformed into the depth image frame $(x'-y'-z')$ via Eq. (2) and then projected into the depth image according to its spherical coordinate $(\varphi, \theta, d)$ and image resolutions $(r_h, r_v)$. **b** A depth image is constructed using points sampled in a fixed interval $T$.

accumulating a frame (i.e., an online system). This point-by-point detection mechanism enables an extremely low detection latency (i.e., 2-4 $\mu$s per point). In contrast, existing methods all operate on frames (e.g., 10 Hz), where a current frame of LiDAR points (or even a future frame[19]) must be accumulated to compare the point cloud difference[16], compute point descriptors[17], extract surfels[18] or semantic features[24,25], or apply a neural network classifier[26–35]. This point accumulation completely relinquishes the high-rate sampling nature of a LiDAR sensor and causes hundreds of milliseconds delay.

The first occlusion principle employed by M-detector to detect objects crossing the LiDAR laser rays (Fig. 3a) is not the first time it is used for moving object segmentation. Ref. 56, 57 used the first occlusion principle to detect scene changes in long-term simultaneous localization and mapping (SLAM) and ref. 58 used a similar occlusion principle for environment monitoring. In these applications, the two frames for movement detection were often temporally and spatially distant. Moreover, they were not online systems and required either all[56] or one future frame[57] or a prior map of the environment[58] for the detection. In contrast, M-detector exploits the occlusion principle to detect moving objects continuously and in an online and real-time fashion without any prior map of the scene. Further, the second and third occlusion principles proposed in this work for detecting objects moving along the LiDAR laser rays (Fig. 3b, c) have rarely been reported before to the best of our knowledge.

## Methods

As shown in Fig. 4, the key step of M-detector is event detection (Fig. 4b), which examines the occlusion between the current point and points in the past. The event detection consists of three independent tests, which correspond to the three occlusion principles (Fig. 3), respectively. If any of the tests are positive, the current point will be labeled as an event or else non-event. In all the three tests, the occlusion check with previous points is achieved by organizing previous points into a series of depth images. Moreover, after each occlusion check in the tests, map consistency check is applied to reject any false occlusion if the point is too close to any stationary map points. Finally, points are accumulated into a frame where clustering & region growth is utilized to reject the isolated event points and bring back non-event points mislabeled by the three tests.

### Depth image

A depth image refers to a two-dimensional array, where each location (i.e., a pixel) saves the information, such as event labels and spherical coordinates of each point that was projected to the reception field of this pixel (Fig. 9a). A pixel also summarizes those statistical information, including the number of points in this pixel, the maximum depth, and the minimum depth among all the contained points for fast inquiry. A depth image is attached with a pose $(\mathbf{R}, \mathbf{t})$, referred to as the

depth image pose, which indicates the pose of the LiDAR body frame $(x'-y'-z')$ with respect to a global reference frame $(x-y-z)$, at the moment the depth image is constructed.

**Depth image construction.** Depth images are constructed in a fixed interval $T$, where LiDAR points sampled in this interval are all used to construct the image (Fig. 9b). We choose the LiDAR pose at the beginning of the interval as the depth image pose, so that all the subsequent points in the interval can be projected to the depth image with their event labels. Specifically, let $^G\mathbf{p}$ be a labeled point, which has been registered to the global reference frame (e.g., by the odometry module). The projection of the point to the depth image could be achieved by the following steps. First, transform the point into the depth image frame via

$$^L\mathbf{p} = \mathbf{R}^{-1}(^G\mathbf{p} - \mathbf{t}) \tag{2}$$

where $^L\mathbf{p} = [^Lp_x, {}^Lp_y, {}^Lp_z]^T$ denotes the coordinate of the point in the depth image frame. Second, project the transformed point to the image pixels by obtaining its spherical coordinates (Fig. 9a):

$$\varphi = \text{atan2}\left(^Lp_y, {}^Lp_x\right) \tag{3}$$

$$\theta = \text{atan2}\left(\sqrt{^Lp_x^2 + {}^Lp_y^2}, {}^Lp_z\right) \tag{4}$$

$$d = \sqrt{^Lp_x^2 + {}^Lp_y^2 + {}^Lp_z^2} \tag{5}$$

where $\varphi$, $\theta$, and $d$ represent the point's azimuthal angle, polar angle, and radial distance (i.e., depth) in spherical coordinates, respectively. Finally, determining the pixel location $(i, j)$ of the projected point by

$$i = \lfloor(\varphi + \pi)/r_h\rfloor, \quad j = \lfloor(\theta + \pi/2)/r_v\rfloor \tag{6}$$

where $r_h$, $r_v$ are the horizontal, and vertical pixel size (i.e., resolution) of the depth image, and $\lfloor . \rfloor$ denotes the floor function. The depth image could be implemented at the fixed resolution mentioned here or at multiple resolutions if necessary. After the pixel location is determined, the point information, such as the spherical coordinate and event labels, are saved to the pixel and the pixel statistical information (e.g., point number, maximum and minimum depths) are updated accordingly.

Note that the construction of the current depth image $\mathbf{I}_k$ is parallel to the event detection (Fig. 4a), which depends on only depth images up $\mathbf{I}_{k-1}$ (Fig. 4b). Therefore, the construction does not cause any detection delay for the event detection in both point-out and frame-out modes. Furthermore, an initialization stage is required to

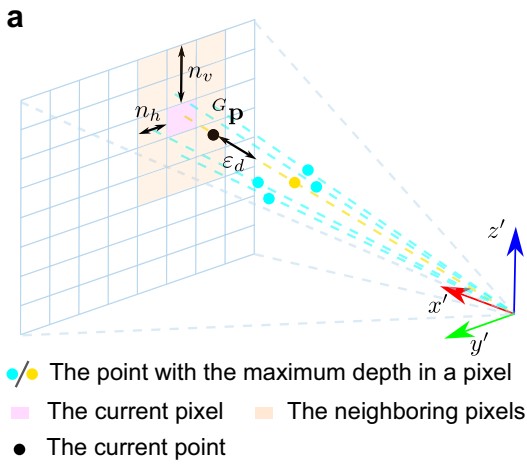

**a** The point with the maximum depth in a pixel
The current pixel    The neighboring pixels
The current point

**b** The point with the minimum depth in a pixel
The current pixel    The neighboring pixels
The current point

**Fig. 10 | Illustration for occlusion check in M-detector. a** In this figure, after projecting the current point $^G\mathbf{p}$ into the depth image $(x' - y' - z')$ according to Eqs. (2)–(6), the depth of $^G\mathbf{p}$ (denoted in black) is larger (with threshold $\varepsilon_d$) than the maximum depth saved in current pixel (denoted in yellow) and that in the neighboring pixels (denoted in cyan), hence the point is considered as being occluded by

points in this depth image. **b** In this figure, after projecting the current point $^G\mathbf{p}$ into the depth image $(x' - y' - z')$ according to Eqs. (2)–(6), the depth of $^G\mathbf{p}$ (denoted in black) is smaller (with threshold $\varepsilon_d$) than the minimum depth saved in current pixel (denoted in purple) and that in the neighboring pixels (denoted in orange), hence the point is considered as occluding points in this depth image.

construct the first few depth images, during which the event detection is not performed. This initialization time is often small (e.g., 0.5 s).

**Occlusion check on depth image.** Based on the depth image representation of the previous points, the occlusion check between the current point and previous points in a depth image can be conducted as follows. Let $^G\mathbf{p}$ be the current point to check, it is projected into the depth image according to Eqs. (2)–(6) to obtain its pixel location and depth. If the point depth is larger (with threshold $\varepsilon_d$) than the maximum depth saved in current pixel and that in the neighboring pixels, the point is considered as being occluded by points in the depth image (Fig. 10a). If the point depth is smaller (with threshold $\varepsilon_d$) than the minimum depth saved in current pixel and that in the neighboring pixels, the point is considered as occluding points in the depth image (Fig. 10b). Otherwise, the occlusion between the current point and points in the depth image is undetermined. The neighboring pixels are those pixels whose distance to the current pixel is less than $n_h$ pixels in the horizontal direction and $n_v$ in the vertical direction.

The above occlusion check based on the maximum and minimum depth of a pixel avoids the enumeration of each point in the pixel, leading to a high computation efficiency. However, it does not give all points in the depth image that occlude or are occluded by the current point. To address this issue, an alternative occlusion check is also developed. In this strategy, every point in the current pixel and neighboring pixels is enumerated. If its position on the image plane is within a small neighborhood ($\varepsilon_h$ degrees in the horizontal direction and $\varepsilon_v$ degrees in the vertical direction) of the current point, it is appended to the neighboring point set $\mathcal{N}$. Then, we compare the depth of each point in the set $\mathcal{N}$ to the current point. If the point is $\varepsilon_d$ further, it is added to the set of occluded points $\mathcal{N}_{occluded}$. Otherwise, if the point is $\varepsilon_d$ closer, it is added to the set of occluding points $\mathcal{N}_{occluding}$. This finer occlusion check will be used in the test two and test three of the event detection, while the previous occlusion check based on pixels will be used in the test one (Fig. 4 and the following section).

**Event detection**

The event detection conducts three independent tests according to the three occlusion principles for each current point (Fig. 3). If any of them is positive, the point is labeled as an event.

**Event detection for perpendicular movement.** The first test detects event points of objects whose moving direction is perpendicular to the LiDAR laser rays. In this case, the object must occlude the background objects that have been observed previously (Fig. 3a). According to this occlusion principle, the test is developed as shown in Fig. 4b. We project the current point $\mathbf{p}^{curr}$ to the most recent $N$ depth images and examine its occlusion with each of them. If there are more than $M_1$ depth images ($M_1 \leq N$) whose points are occluded by the current point, the current point is labeled as an event.

Event labeling based solely on the occlusion check may cause some false positives due to the limited resolution of the depth image. For example, for a stationary object placed in front of a stationary background, points collected on the object edge may be projected to the pixel with points from the background due to the rounding error in the projection (Eq. (6)). Consequently, these points will be falsely labeled as events. Other factors, such as measurement noises, could also bring some false positive detection.

To mitigate such false positive detection, we introduce a rejection strategy termed as map consistency check. The key idea is that an event point should not lie in the neighborhood of any stationary map points. Otherwise, it could be safely treated as part of the stationary objects since it cannot distinguish if this occlusion is caused by true movements or measurement noises. If the map consistency check is positive (i.e., the point is close to any map points), the candidate event point will be rejected and labeled as a non-event.

**Event detection for parallel movement.** The second test detects points of objects that are moving away from the LiDAR in parallel to the sensor laser rays. In this case, objects must be occluded by themselves repeatedly (Fig. 3b). According to this occlusion principle, the test is developed as shown in Fig. 4b. We examine if the current point $\mathbf{p}^{curr}$ is occluded by any points in all of the previous $M_2$ depth images (denoted as $\mathbf{p}^{\mathbf{l}_{k-1}}, \cdots, \mathbf{p}^{\mathbf{l}_{k-M_2}}$, respectively). Further, we test if each of these points is occluded by all of its subsequent points (i.e., if $\mathbf{p}^{\mathbf{l}_{k-i}}$ is occluded by $\mathbf{p}^{\mathbf{l}_{k-j}}$ for all $i = 1, \cdots, M_2 - 1$ and $j = i + 1, \cdots, M_2$). If all these tests are positive, it implies that the current point $\mathbf{p}^{curr}$ and previous points $\mathbf{p}^{\mathbf{l}_{k-1}}, \cdots, \mathbf{p}^{\mathbf{l}_{k-M_2}}$ are on objects that are moving in the mentioned way, and hence should be labeled as events. To reject possible false positives, we further apply the map consistency check similar to test one to $\mathbf{p}^{curr}$ and $\mathbf{p}^{\mathbf{l}_{k-1}}, \cdots, \mathbf{p}^{\mathbf{l}_{k-M_2}}$ after each of the above occlusion test. If all the occlusion tests are positive without being rejected by their respective

map consistency check, the current point is labeled as an event. Otherwise, it is a non-event point.

The third test detects points of objects that are moving towards the LiDAR in parallel to the sensor laser rays. In this case, objects must occlude themselves repeatedly (Fig. 3c). According to this occlusion principle, the test is developed as shown in Fig. 4b. We examine if the current point $\mathbf{p}^{curr}$ occludes any points in all of the previous $M_3$ depth images (denoted as $\mathbf{p}^{l_{k-1}}, \cdots, \mathbf{p}^{l_{k-M_3}}$, respectively). Further, we test if each of these points occludes all of its subsequent points (i.e., if $\mathbf{p}^{l_{k-i}}$ occludes $\mathbf{p}^{l_{k-j}}$ for all $i = 1, \cdots, M_3-1$ and $j = i+1, \cdots, M_3$). If all these tests are positive, it implies that the current point $\mathbf{p}^{curr}$ and previous points $\mathbf{p}^{l_{k-1}}, \cdots, \mathbf{p}^{l_{k-M_3}}$ are on objects that are moving in the mentioned way, and hence should be labeled as events. Similarly, we further apply the map consistency check to $\mathbf{p}^{curr}$ and $\mathbf{p}^{l_{k-1}}, \cdots, \mathbf{p}^{l_{k-M_3}}$ after each of the above occlusion test. If all the occlusion tests are positive without being rejected by their respective map consistency check, the current point is labeled as an event. Otherwise, it is a non-event point.

**Map consistency check.** Map consistency check aims to reject wrong event points according to the principle that a point on the moving object should be distant from stationary map points. To maximally re-use the existing data structure, we perform the map consistency check on the depth images, where the map points have been saved. Since one depth image contains only a part of map points, we perform map consistency check on multiple recent depth images along with each occlusion check required in the three tests above. Specifically, after each occlusion check, we retrieve all non-event points in the pixel where the current point is projected and that in neighboring pixels. For any retrieved point, if its spherical coordinate is in a small neighborhood (i.e., $[-\varepsilon_\varphi, \varepsilon_\varphi]$ degrees in azimuthal angle, $[-\varepsilon_\theta, \varepsilon_\theta]$ degrees in polar angle, and $[-\varepsilon_f, \varepsilon_b]$ meters in depth) of the current point, the current point is considered as a stationary point and the occlusion check on this depth image (if positive) should be rejected.

To further increase the accuracy of the map consistency check, we interpolate the depth at the current point image location, using the neighboring points retrieved above. Specifically, given a depth image and an image location $(\varphi, \theta) \in \mathbb{R}^2$ whose depth needs to interpolate, we first find its three neighboring points in the depth image, $\mathbf{p}_i$ ($i = 1, 2, 3$) with spherical coordinates $(\varphi_i, \theta_i, d_i)$. Then, we determine the contribution of each point, $w_i$, by solving the following three linear equations:

$$\sum_{i=1}^{3}\left(\begin{pmatrix} \varphi_i \\ \theta_i \end{pmatrix} w_i\right) = \begin{pmatrix} \varphi \\ \theta \end{pmatrix}, \qquad \sum_{i=1}^{3} w_i = 1 \qquad (7)$$

With the solved weights $w_i$, the interpolated depth is hence

$$\bar{d} = \sum_{i=1}^{3} d_i w_i \qquad (8)$$

To avoid extrapolation causing degraded depth estimate, the three neighboring points should be close to $(\varphi, \theta)$ as much as possible, and the $(\varphi, \theta)$ should lie in the convex hull formed by $(\varphi_i, \theta_i)$ (i.e., $w_i \geq 0$, $i = 1, 2, 3$) (Supplementary Fig. 2). To fulfill these two conditions, we enumerate all the neighboring points in a neighbor of $\varepsilon_h \times \varepsilon_v$ degrees around $(\varphi, \theta)$. First, all $3n$ combinations of three points among all the $n$ neighboring points are listed and sorted based on the sum of the absolute difference between the $(\varphi, \theta)$ values and the $(\varphi_i, \theta_i)$, denoted as $\sum_{i=1}^{3} |\varphi - \varphi_i| + |\theta - \theta_i|$. Then, the weights $w_i$ are solved using Eq. (7) in the order of the sorted sums (from small to large). If the resulting weights satisfy the condition of $w_i \geq 0$, $i = 1, 2, 3$, the interpolated position is considered to fall within the convex hull formed by the three points and the interpolated depth is obtained from Eq. (8). If the condition is not satisfied, the process continues with the next

combination of three points. If no successful interpolation is achieved among all neighboring points, the depth interpolation fails.

If the depth is interpolated successfully and its value is in a small neighborhood (i.e., within $[-\varepsilon_f, \varepsilon_b]$ meters) of the current point, the current point is considered as a stationary point. Since the interpolation is more time-consuming than the direct comparison, it is only performed for points at far, where the point density is low.

### Clustering and region growth
Clustering and region growth are designed to further suppress possible false positive event points in the current depth image. The consequences are twofold. In the point-out mode, they do not improve the detection accuracy of points in the current frame (since these event labels have been output), but they prevent the adverse effect of current false labels on future point detection. For example, the map consistency check for future points will require to excluding current event points, hence being affected by the current labels. In the frame-out mode, the clustering and region growth further improve the event detection accuracy of the current frame.

Clustering is performed to remove isolated event points since it is unlikely to collect only one point on a moving object with existing LiDARs' resolutions. Clustering directly on event points could be computationally expensive when the point density is high. To address this issue, event points are voxelized (voxel size $L_v$). Those voxels containing event points are called event voxels, which are clustered by DBSCAN[59] based on their center position. In this way, isolated event voxels and their contained event points are rejected. Furthermore, all raw points in event voxels are labeled as events, which helps bring back some event points that were mislabeled by the event detection as non-events.

The above voxelization can recall some event points, but the number is not high due to the small voxel size (e.g., 0.3 m), which is necessary to suppress false detection. To recover more event points, region growth is performed. For each clustered group of event voxels in above steps, the minimum axis-aligned bounding box (AABB) containing all the event voxels in this cluster is extracted. Then, the AABB is expanded to twice its size resulting in an expansion space. A ground plane is fitted using points in the expansion space by RANSAC[60] so that the ground points and the respective voxels in the expansion space and AABB can be removed. Then, we grow the event voxels in the AABB by examining its neighboring voxels recursively. If its neighboring voxels contain non-ground points, they are merged to the set of event voxels. The growth ends until reaching the boundary of the expansion space. Finally, all raw points in the set of event voxels are labeled as events and used to construct the depth image.

### Reporting summary
Further information on research design is available in the Nature Portfolio Reporting Summary linked to this article.

## Data availability
The source data presented in the manuscript have been deposited in Figshare and can be accessed at https://doi.org/10.6084/m9.figshare.24481966.

## Code availability
Source code of M-detector has been provided on the GitHub repository. This code is freely accessible. All the information needed to install and use it, as well as any updates, can be found here: https://github.com/hku-mars/M-detector.

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

## Acknowledgements

This project has received funding from Hong Kong Research Grants Council General Research Fund (17206421) and DJI research donation fund, both received by F. Z. We thank Nan Chen and Jiarong Lin, who offered valuable suggestions to the manuscript. We sincerely appreciate Yunfang Ren, Chaoran Yu, Hongyi Pan, Yixi Cai, Guozheng Lu, Fangcheng Zhu, Xiyuan Liu, and Erchao Rong for their help on experiments and data acquisition. Furthermore, we are truly grateful to Dr. Ximin Lyu for providing the space for indoor experiments. At last, we also give thanks to Dr. Xieyuanli Chen, the author of LMNet[30], for his discussion with us on the time consumption of LMNet.

## Author contributions

F.Z. proposed the initial idea of the research. With the advice of F.Z., H.W., Y.L., and W.X. designed the complete system framework and experiments. H.W. and Y.L. implemented all software modules. X.W. validated part of the feasibility of the initial idea and conducted the initial analysis. With the contribution of F.K., H.W. performed the UAV experiments and data acquisition of *AVIA-Indoor*. Y.L. and H.W. collected the data demanded by the five applications in Results. H.W., Y.L., and F.Z. finished all data analyses for different datasets and wrote the manuscript. F.Z. provided funding and supervised the research.

## Competing interests

The University of Hong Kong filed a provisional U.S. patent application (No. PCT/CN2023/085922) on this work on 4 April 2022. F.Z., W.X., and H.W. are inventors. All the other authors declare no competing interests.
