## [Peer Review File · Nature Communications]

Moving Event Detection from LiDAR Point StreamsREVIEWER COMMENTS

Reviewer #1 (Remarks to the Author):

The authors have demonstrated a method for moving event detection by using high acquisition-rate LiDAR. The feature that immediately detects each point movement upon its arrival makes this method exceed the traditional frame-based scheme on decrease detection latency and hardware requirements. The experiments on using LiDAR for moving detection is interesting and the technical improvement is obvious. I think this manuscript can be accepted before the following concerns have been addressed.

1. Since the moving detection is based on LiDAR, two detections of the moving point are required to compare and judge the movement. Does the author detect the same point two times before detecting the next point?
2. I suggest the authors add more explanation and description of the M-detector workflow for consideration of a broad audience.
3. It seems that the point stream density has an upper limit due to the detection latency per point and frame period. Can you give a comment on this upper limit and its application preference? Does the detection latency still have room for improvement?
4. I suggest the authors add the evaluation of point stream density improvement to demonstrate the advantages of the M-detector method since high imaging resolution is attractive for autonomous driving.
5. According to the results shown in Fig.5(a), why the IoU is different for point-out and frame-out modes? Which mode is more attractive for practical use?
6. Does the FMCW LiDAR have a performance gain on moving detection, considering its 4D imaging (simultaneous depth and velocity detection) ability?
7. The subtitles for each section should be renamed to make the content more attractive.
8. What's the constraint of the safety threshold of object speed for obstacle avoidance? How to improve this threshold?

Reviewer #2 (Remarks to the Author):

The authors have proposed and demonstrated an interesting approach to reducing latency of moving event detection using LiDAR. The authors introduce three distinct occlusion tests to distinguish between an object moving perpendicular to the rays of the LiDAR, moving away from the LiDAR, and moving towards the LiDAR. The proposed scheme is then compared with other approaches to event detection. Specifically, consistency check, occupancy map, semantic segmentation, and motion segmentation methods are used as a baseline with which the novel scheme is compared. The occlusion-based approach does seem to enable event detection with significantly lower latency than the other approaches tested however there are some points that I think should be made clear before the paper is published.

While the enhancement in moving event detection latency is impressive, there are other approaches to moving object detection with LiDAR that are not mentioned. Doppler LiDAR uses the frequency shift resulting from the doppler shift of the LiDAR laser beam upon reflection from a moving target. By measuring the frequency difference between the outgoing and incoming light, one can---in real time---measure the velocity of the object from which the light was reflected.

-The authors should provide a more comprehensive list of moving event detection protocols from outside the computer vision community such as Doppler shift LiDAR.

In the introductory section, the authors introduce consistency check, occupancy map, semantic segmentation, and motion segmentation methods for moving object segmentation and explain in what ways they fall short. However, in the results section, the M-detector approach is only compared with LMNet and SMOS models.

-The authors should explain in the main manuscript how the LMNet and SMOS models correspond to the four approaches mentioned in the introduction. If any of the four approaches mentioned in the introduction is not used as a benchmark for the results, this should be stated and a justification for why it was omitted should be given.

Reviewer #3 (Remarks to the Author):

This paper presents a rapid-reaction moving object detection method. The overall contribution is significant since the detection latency is reduced from the rotation frequency of LiDARs around 0.1s to laser beam emission frequency at 0.01s levels. Such improvement in speed will revolutionize LiDAR-based object detection in many applications, as illustrated by the authors' videos. Here are the major comments.

1. Threshold-based occlusion detection: Lines 549-554, Page 14. The authors used a fixed threshold for neighborhood determination. Such setup may restrict the application of the algorithm in spaces, such as intersections, where vehicles can come at many different angles. Then the threshold degrees may be difficult to calibrate to cover all directions.

1. Object segmentation: Lines 663-685, Pages 17-18. DBSCAN and RANSAC are used in clustering scattered point events into objects. Given the high-computational cost of both algorithms, the authors may want to provide partial computational time measurements of both processes in the overall M-detector processor's computational time. Were they the main source of delay for the M-detector algorithm? If not, what type of optimization of those algorithms is conducted?

2. Object tracking: The authors do not detect an object even if it is previously detected and pause its motion in a couple of seconds, which can be a major flaw in real-world applications such as collision avoidance in congested stop-and-go traffic or at traffic signals. Such flaws may be caused by their algorithm not including any tracking capability. If an object can be tracked, whether or not they become static will not be an issue. The authors may want to explore some complementary LiDAR tracking algorithms to address this issue.

3. Model evaluation and calibration, Equation 1, Page 5: Since this method is a standard model-based method that involves the calibration of thresholds, the author should provide some threshold

calibration results, such as the precision-recall curves or receiver operating characteristic curves. Some guidelines should be provided on how to calibrate those thresholds, such as neighborhood or occlusion thresholds, including its the feasible range, optimization methods, and the calibrated values for each sample application.

4. Other minor comments:

Line 523 and 537, Page 14: "depth" is never clearly defined through equations 2-6. It kept being referred to without further details. How was depth calculated for a point? Which coordinate system is it in? Is it a vector with three elements? Is d in Equation 5 the depth?

Lines 257-260, Page 7: Please provide the processing time of the M-detector also in microseconds, not just relative percentages of the computational time of LMNet and other comparison algorithms.

Response to Reviewer Comments

August 8, 2023

We greatly appreciate all the helpful comments, and have done our best to carefully address them. Below, [Q] is used to indicate questions or comments from the reviewer, and [Reply:] is used to indicate our corresponding answers. Please note that cited texts from our manuscript are presented in blue respectively. The reviewer's comments have been listed in order below in the italic text, followed by our responses in normal text.

REVIEWER COMMENTS

Reviewer #1 (Remarks to the Author):

The authors have demonstrated a method for moving event detection by using high acquisition-rate LiDAR. The feature that immediately detects each point movement upon its arrival makes this method exceed the traditional frame-based scheme on decrease detection latency and hardware requirements. The experiments on using LiDAR for moving detection is interesting and the technical improvement is obvious. I think this manuscript can be accepted before the following concerns have been addressed.

1. Since the moving detection is based on LiDAR, two detections of the moving point are required to compare and judge the movement. Does the author detect the same point two times before detecting the next point?

2. I suggest the authors add more explanation and description of the M-detector workflow for consideration of a broad audience.

3. It seems that the point stream density has an upper limit due to the detection latency per point and frame period. Can you give a comment on this upper limit and its application preference? Does the detection latency still have room for improvement?

4. I suggest the authors add the evaluation of point stream density improvement to demonstrate the advantages of the M-detector method since high imaging resolution is attractive for autonomous driving.

5. According to the results shown in Fig.5(a), why the IoU is different for point-out and frame-out modes? Which mode is more attractive for practical use?
6. Does the FMCW LiDAR have a performance gain on moving detection, considering its 4D imaging (simultaneous depth and velocity detection) ability?
7. The subtitles for each section should be renamed to make the content more attractive.
8. What's the constraint of the safety threshold of object speed for obstacle avoidance? How to improve this threshold?

Reviewer #2 (Remarks to the Author):

The authors have proposed and demonstrated an interesting approach to reducing latency of moving event detection using LiDAR. The authors introduce three distinct occlusion tests to distinguish between an object moving perpendicular to the rays of the LiDAR, moving away from the LiDAR, and moving towards the LiDAR. The proposed scheme is then compared with other approaches to event detection. Specifically, consistency check, occupancy map, semantic segmentation, and motion segmentation methods are used as a baseline with which the novel scheme is compared. The occlusion-based approach does seem to enable event detection with significantly lower latency than the other approaches tested however there are some points that I think should be made clear before the paper is published.

While the enhancement in moving event detection latency is impressive, there are other approaches to moving object detection with LiDAR that are not mentioned. Doppler LiDAR uses the frequency shift resulting from the doppler shift of the LiDAR laser beam upon reflection from a moving target. By measuring the frequency difference between the outgoing and incoming light, one can—in real time—measure the velocity of the object from which the light was reflected.

-The authors should provide a more comprehensive list of moving event detection protocols from outside the computer vision community such as Doppler shift LiDAR.

In the introductory section, the authors introduce consistency check, occupancy map, semantic segmentation, and motion segmentation methods for moving object segmentation and explain in what ways they fall short. However, in the results section, the M-detector approach is only compared with LMNet and SMOS models.

-The authors should explain in the main manuscript how the LMNet and SMOS models correspond to the four approaches mentioned in the introduction. If any of the four approaches mentioned in the introduction is not used as a benchmark for the results, this should be stated and a justification for why it was omitted should be given.

Reviewer #3 (Remarks to the Author):

This paper presents a rapid-reaction moving object detection method. The overall contribution is significant since the detection latency is reduced from the rotation frequency of LiDARs around 0.1s to laser beam emission frequency at 0.01s levels. Such improvement in speed will revolutionize LiDAR-based object detection in many applications, as illustrated by the authors' videos. Here are the major comments.

1. Threshold-based occlusion detection: Lines 549-554, Page 14. The authors used a fixed threshold for neighborhood determination. Such setup may restrict the application of the algorithm in spaces, such as intersections, where vehicles can come at many different angles. Then the threshold degrees may be difficult to calibrate to cover all directions.

2. Object segmentation: Lines 663-685, Pages 17-18. DBSCAN and RANSAC are used in clustering scattered point events into objects. Given the high-computational cost of both algorithms, the authors may want to provide partial computational time measurements of both processes in the overall M-detector processor's computational time. Were they the main source of delay for the M-detector algorithm? If not, what type of optimization of those algorithms is conducted?

3. Object tracking: The authors do not detect an object even if it is previously detected and pause its motion in a couple of seconds, which can be a major flaw in real-world applications such as collision avoidance in congested stop-and-go traffic or at traffic signals. Such flaws may be caused by their algorithm not including any tracking capability. If an object can be tracked, whether or not they become static will not be an issue. The authors may want to explore some complementary LiDAR tracking algorithms to address this issue.

4. Model evaluation and calibration, Equation 1, Page 5: Since this method is a standard model-based method that involves the calibration of thresholds, the author should provide some threshold calibration results, such as the precision-recall curves or receiver operating characteristic curves. Some guidelines should be provided on how to calibrate those thresholds, such as neighborhood or occlusion thresholds, including its the feasible range, optimization methods, and the calibrated values for each sample application.

5. Other minor comments: Line 523 and 537, Page 14: "depth" is never clearly defined through equations 2-6. It kept being referred to without further details. How was depth calculated for a point? Which coordinate system is it in? Is it a vector with three elements? Is d in Equation 5 the depth?

Lines 257-260, Page 7: Please provide the processing time of the M-detector also in microseconds, not just relative percentages of the computational time of LMNet and other comparison algorithms.

Point-by-point Responses to the Reviewers' Comments

Reviewer 1

Q 1 *The authors have demonstrated a method for moving event detection by using high acquisition-rate LiDAR. The feature that immediately detects each point movement upon its arrival makes this method exceed the traditional frame-based scheme on decrease detection latency and hardware requirements. The experiments on using LiDAR for moving detection is interesting and the technical improvement is obvious. I think this manuscript can be accepted before the following concerns have been addressed.*

Reply: We are grateful to the reviewer for your compliment on our work. We carefully considered the reviewer's comments and made the necessary revisions to further improve our work. We hope this response and the revised manuscript satisfactorily address the reviewer's comments.

Q 2 *Since the moving detection is based on LiDAR, two detections of the moving point are required to compare and judge the movement. Does the author detect the same point two times before detection?*

Reply: Thank you for the reviewer's comments. We appreciate your interests in our work and the valuable feedback on our methodology. In response to this question, we would like to clarify that M-detector does not detect the same point twice before detection. Factually, due to the potentially non-repetitive scanning pattern of a LiDAR (e.g., Livox LiDARs) or the irregular ego-motion, it is difficult to scan the same point at different times or in different frames. Therefore, it is not possible to compare and judge the movement of the same point. M-detector addresses this issue by comparing the current point with previously sampled points in a small neighborhood on the depth image. If the current point and previous points in the neighborhood exhibit a certain occlusion relation (as detailed in Fig. 4b), it is determined as an event point.

Q 3 *I suggest the authors add more explanation and description of the M-detector workflow for consideration of a broad audience.*

Reply: We thank the reviewer for the helpful suggestion. We agree that more explanation and description of the M-detector workflow will help clarify our methodology for moving event detection from LiDAR point streams better. Following the reviewer's suggestion, we have added a detailed explanation of M-detector's workflow in Supplementary Section S2 titled "*Detailed Explanation for M-detector's Workflow*". This added section is introduced

according to Fig. 4, which provides a step-by-step demonstration of how M-detector works to determine the points whether are event or non-event ones from its input to output.

Q 4 *It seems that the point stream density has an upper limit due to the detection latency per point and frame period. Can you give a comment on this upper limit and its application preference? Does the detection latency still have room for improvement?*

Reply: We thank the reviewer for the insightful comments. The reviewer is correct that there is an upper limit on the number of points that can be processed by M-detector in its current implementation. This upper limit is determined by T_f/t_p , where T_f denotes the frame period of a LiDAR frame, and t_p denotes the average processing time per point.

The upper limit in point stream density of M-detector is depicted in Fig. R1. As can be seen, M-detector can process up to 167,845 per frame at 10Hz without affecting its real-time capability on a moderate computer with a central processing unit (CPU) of Intel i7-10700 (2.90 GHz, 8 cores) and 48 GB random access memory (RAM). This point density upper limit allows it to be used with most of existing LiDARs, including Velodyne HDL-64E (130,000 points per frame), Waymo-64 (160,000 points per frame), Velodyne HDL-32E (80,000 points per frame), and Livox AVIA (24,000 points per frame), that were widely used in applications including autonomous driving, UAV obstacle avoidance, traffic monitoring, surveillance and people counting, mapping, etc. Results in the section of *Application on autonomous driving* to *Application on mapping* also verified the potential of M-detector in these applications.

Regarding the reviewer’s second comment, he/she is absolutely right that there is room for improvement in the detection latency and time consumption of M-detector. The most obvious method is to fully leverage parallel computing afforded by GPUs or CPUs with more cores or threads into our procedure, which can lower the average processing time per point (i.e., t_p) by two means. For one thing, M-detector detects a point through tests one to three in turn in the current implementation. The processing time of event detection for each point is the sum of the processing time of the three tests. If these three tests can be processed in parallel, the latency will be decreased to the maximal, not the total, processing time of the three tests. For another, the depth image construction is performed sequentially for each point currently. Parallel computing can be applied here to lower the processing time of depth image construction.

We have added the above discussion about point density upper limit and improving methods in Supplementary Section S1.2 of the revised manuscript.

Q 5 *I suggest the authors add the evaluation of point stream density improvement to demonstrate the advantages of the M-detector method since high imaging resolution is attractive for autonomous driving.*

Reply: We appreciate the reviewer’s constructive suggestion. Following the reviewer’s suggestion about point density improvement of our method against other existing methods, we have predicted the point density upper limits of our proposed M-detector against other state-of-the-art methods including SMOS and LMNet, by linearly fitting the time consumption on existing datasets.

As illustrated in Fig. R1, the circle points in different colors denote the time consumption of the corresponding method on that dataset with certain point stream density. The upper limits denoted by stars are obtained by linearly fitting time consumption on existing datasets. For LMNet-1 and LMNet-8*, if they are performed on CPU, no valid upper limits can be fitted from existing data. If they are performed on GPU, their upper limits are 59,336 and 5,657 respectively. For SMOS, its upper limit is 6,734. As for M-detector, the upper limit is 167,845, which is far greater than the other methods. This significant improvement enables our method to run in real-time with a wider choice of LiDAR sensors and limited onboard computing power. Figure R1 and the corresponding analysis have been added to Supplementary Section S1.2 of the revised manuscript.

Figure R1: **Per-frame time consumption of different methods versus the point number in a frame.** Each circle point in different colors denotes the time consumption of the corresponding method on that dataset with certain point numbers. These data are the same as the values tested in the section of *Time consumption and detection latency*. The upper limits (i.e., the star symbol) of different methods are obtained by linearly fitting the data of each method. Note that point number per frame is shown in the linear scale while time consumption per frame is shown in the log scale. The time is evaluated on a moderate computer with a central processing unit (CPU) of Intel i7-10700 (2.90 GHz, 8 cores) and 48 GB random access memory (RAM).

Q 6 According to the results shown in Fig.5(a), why the IoU is different for point-out and frame-out modes? Which mode is more attractive for practical use?

Reply: We thank the reviewer for raising this point. The main difference between point-out mode and frame-out mode is caused by the procedure of clustering and region growth. In point-out mode, M-detector outputs the labels immediately after the event detection module. In contrast, in frame-out mode, after event detection, points are accumulated for one period for clustering and region growth. This step can utilize the information of not only the current individual point but also all other points in the current frame, leading to the rejection of isolated event points and recall of more event points mislabeled in the event detection. The additional clustering and region growth steps often increase the accuracy but introduce a longer delay due to the time required for point accumulation, clustering, and region growth.

The choice of mode is determined by the specific application requirements. For example, for applications where low latency (i.e., several milliseconds) is a must, such as avoidance of fast-moving objects for UAVs, point-out mode is preferred. In contrast, for applications that can tolerate longer latency (i.e., tens of microseconds), such as mapping and traffic monitoring, frame-out mode is more preferable due to the potential gain in accuracy.

Q 7 Does the FMCW LiDAR have a performance gain on moving detection, considering its 4D imaging (simultaneous depth and velocity detection) ability?

Reply: We thank the reviewer for the inspiring comments. The reviewer is correct that FMCW LiDAR leads to a performance gain for M-detector due to the additional Doppler-velocity measurements along the laser ray. The velocity measurement provides effective prior information for test two and test three of M-detector. However, for test one of M-detector, which detects movements perpendicular to the laser ray, it does not benefit from the FMCW LiDAR since no Doppler-velocity can be measured at this direction.

Compared with FMCW LiDAR that measures the radial velocity based on the Doppler effect, M-detector is designed based on ToF LiDAR and can detect movement both parallel and perpendicular to the laser ray, leading to a more unified and comprehensive solution. Furthermore, although FMCW LiDAR has the advantage of measuring depth and velocity simultaneously, FMCW LiDAR's scene acquisition time could take several times longer than LiDAR's [1], which restricts its frame rate that is crucial for robotics applications. Moreover, the current FMCW system requires a fairly large processing unit, leading to a system of bigger size, weight, and power (SWaP) requirements [2].

Following the reviewer's suggestion, we have added the review of FMCW LiDAR to the Section *Introduction* of the revised manuscript, cited as below:

“Moving events detection could be achieved at the measuring stage of a LiDAR sensor, such as the Frequency-Modulated Continuous Wave Laser Detection and Ranging (FMCW-

LADAR) sensors. Compared to the standard ToF LiDAR, FMCW-LADAR involves a continuously emitted laser beam and utilizes the Doppler effect to acquire information on range and velocity [13]. While being able to measure the velocity of each measured point, FMCW-LADAR can only measure the point velocity component along a laser ray, failing to detect any movements perpendicular to the ray [14]. Besides, the FMCW-LADAR scene acquisition time could take several times longer than LiDAR's [15], which restricts its frame rate that is crucial for robotics applications. Furthermore, the current FMCW system requires a fairly large processing unit, leading to a system of bigger size, weight, and power (SWaP) requirements [13]."

Q 8 *The subtitles for each section should be renamed to make the content more attractive.*

Reply: We thank the reviewer for this valuable feedback, which has been fully incorporated into the revised manuscript to enhance readability. The changes mainly take place in Section *Methods* and are highlighted accordingly, including: from *Occlusion check* to *Occlusion check on depth image*, from *Test one* to *Event detection for perpendicular movement*, and from *Test two and Test three* to *Event detection for parallel movement*.

Q 9 *What's the constraint of the safety threshold of object speed for obstacle avoidance? How to improve this threshold?*

Reply: Actually we did not set a safety threshold for object speed in the UAV obstacle avoidance experiment. Instead, the safety threshold was set for the object distance from the UAV: once the object distance is less than the safety threshold, the UAV will be triggered to evade the object to avoid a potential collision.

For the constraint of the safety threshold, its maximum value is determined by the sensing range of the LiDAR. As long as the object is scanned by LiDAR, M-detector can detect it, and the UAV can plan to avoid it. The minimum value of the safety threshold is determined by several factors, including 1) the minimum measuring range of the LiDAR l_b . Below l_b , the LiDAR does not measure any point, so the UAV cannot sense the incoming object; 2) the moving speed of the object v_o . If the object moving speed is fast, the safety threshold has to be set larger in order for the UAV to react with sufficient time; and 3) the perception latency t_l . The safety threshold is set to be larger than $\max(l_b, v_o * t_l)$. In general, a suitable safety threshold should be selected between the minimum and maximum values for a specific experiment. If the safety threshold is too large, it may be difficult to accurately predict the obstacle's future position and determine whether it will collide with the UAV. In this case, the UAV may take ineffective evasive action, resulting in a waste of energy. Furthermore, avoiding obstacles from a long distance makes the experimental effect less visually apparent. Conversely, if the safety threshold is too small, collisions may occur due to too tight margin from $\max(l_b, v_o * t_l)$.

In our experiment, we used the Livox AVIA LiDAR, whose maximum sensing range is 230 m, and the minimum sensing range is 3 m. Assuming that the maximum speed of the object is 10 m/s, and the latency of the UAV measured in our experiments is 1.27 ms, then the minimum safety threshold is 3 m and the maximum safety threshold is 230 m. We selected 4 m as the safety threshold.

Reducing the safety threshold can be achieved by using a LiDAR sensor with smaller minimum measuring range. Currently, the most restrictive factor in the experiment is the minimum sensing range of the LiDAR, and thus selecting a LiDAR with a smaller minimum sensing range can effectively reduce the safety threshold.

Reviewer 2

Q 1 *The authors have proposed and demonstrated an interesting approach to reducing the latency of moving event detection using LiDAR. The authors introduce three distinct occlusion tests to distinguish between an object moving perpendicular to the rays of the LiDAR, moving away from the LiDAR, and moving towards the LiDAR. The proposed scheme is then compared with other approaches to event detection. Specifically, consistency check, occupancy map, semantic segmentation, and motion segmentation methods are used as a baseline with which the novel scheme is compared. The occlusion-based approach does seem to enable event detection with significantly lower latency than the other approaches tested however there are some points that I think should be made clear before the paper is published.*

Reply: We appreciate the reviewer’s reviewing efforts and the constructive feedback.

Q 2 *While the enhancement in moving event detection latency is impressive, there are other approaches to moving object detection with LiDAR that are not mentioned. Doppler LiDAR uses the frequency shift resulting from the doppler shift of the LiDAR laser beam upon reflection from a moving target. By measuring the frequency difference between the outgoing and incoming light, one can—in real time—measure the velocity of the object from which the light was reflected. The authors should provide a more comprehensive list of moving event detection protocols from outside the computer vision community such as Doppler shift LiDAR.*

Reply: We thank the reviewer for the valuable feedback. Following the reviewer’s comments, we have added a more comprehensive review of dynamic object detection based on the Doppler LiDAR (we called it frequency-modulated continuous wave laser detection and ranging (FMCW-LADAR)) in the *Introduction* of our manuscript, cited as below:

“Moving events detection could be achieved at the measuring stage of a LiDAR sensor, such as the Frequency-Modulated Continuous Wave Laser Detection and Ranging (FMCW-LADAR) sensors. Compared to the standard ToF LiDAR, FMCW-LADAR involves a continuously emitted laser beam and utilizes the Doppler effect to acquire information on range and velocity [13]. While being able to measure the velocity of each measured point, FMCW-LADAR can only measure the point velocity component along a laser ray, failing to detect any movements perpendicular to the ray [14]. Besides, the FMCW-LADAR scene acquisition time could take several times longer than LiDAR’s [15], which restricts its frame rate that is crucial for robotics applications. Furthermore, the current FMCW system requires a fairly large processing unit, leading to a system of bigger size, weight, and power (SWaP) requirements [13].”

Q 3 *In the introductory section, the authors introduce consistency check, occupancy map, semantic segmentation, and motion segmentation methods for moving object segmentation and explain in what ways they fall short. However, in the results section, the M-detector approach is only compared with LMNet and SMOS models. The authors should explain in the main manuscript how the LMNet and SMOS models correspond to the four approaches mentioned in the introduction. If any of the four approaches mentioned in the introduction is not used as a benchmark for the results, this should be stated and a justification for why it was omitted should be given.*

Reply: We thank the reviewer for the constructive comments. We would like to clarify that LMNet and SMOS are representative methods based on motion segmentation, and occupancy map, respectively. For the other two categories (i.e., semantic segmentation and consistency check), methods based on semantic segmentation only segment movable objects, not moving ones, and thus have a different scope from our method and make a comparison unfair. For methods based on consistency check, to the best of our knowledge, there are no open-source implementations available for a fair comparison. Therefore, there is no benchmark for these two categories of methods.

Following the reviewer’s advice, we have further clarified the categories of LMNet and SMOS at the beginning part of the Section *Results* in the manuscript. The reasons why they are chosen as benchmarks are also provided. They are:

“In all evaluations, we compared the M-detector with two methods, LMNet [30], which is a representative learning-based motion segmentation method, and SMOS [20], a representative method based on occupancy map. Methods based on semantic segmentation and consistency check are not benchmarked. The reason is that methods based on semantic segmentation only segment movable objects, not moving ones, and thus have a different objective from M-detector. For consistency check-based methods, to the best of our knowledge, there are no open-source implementations available for a fair comparison.”

Reviewer 3

Q 1 *This paper presents a rapid-reaction moving object detection method. The overall contribution is significant since the detection latency is reduced from the rotation frequency of LiDARs around 0.1s to laser beam emission frequency at 0.01s levels. Such improvement in speed will revolutionize LiDAR-based object detection in many applications, as illustrated by the authors' videos. Here are the major comments.*

Reply: We are grateful to the reviewer's reviewing efforts and recognition of our work.

Q 2 *Threshold-based occlusion detection: Lines 549-554, Page 14. The authors used a fixed threshold for neighborhood determination. Such setup may restrict the application of the algorithm in spaces, such as intersections, where vehicles can come at many different angles. Then the threshold degrees may be difficult to calibrate to cover all directions.*

Reply: We thank the reviewer for this insightful comment. Factually, a fixed threshold for neighborhood determination will not restrict the application of M-detector in spaces, such as intersections, because the neighborhood threshold is applied on the depth image where a LiDAR sensor often has a uniform resolution regardless of the angular location in the Field of View (FoV). Specifically, as shown in the Fig. R2, the LiDAR's scanning is uniform in all directions, and the depth images used for occlusion check have a 360° horizontal FoV and they are homogeneous in all directions too. Therefore, the size of the neighborhood on the image plane remains the same with a fixed threshold ε_h , no matter at which angle the vehicle appears. This uniform scanning ensures that the detection result is consistent throughout the LiDAR's FoV, regardless of the angular location of the dynamic objects.

Q 3 *Object segmentation: Lines 663-685, Pages 17-18. DBSCAN and RANSAC are used in clustering scattered point events into objects. Given the high-computational cost of both algorithms, the authors may want to provide partial computational time measurements of both processes in the overall M-detector processor's computational time. Were they the main source of delay for the M-detector algorithm? If not, what type of optimization of those algorithms is conducted?*

Reply:

We thank the reviewer for raising the valuable comments. The time consumption of DBSCAN and RANSAC is part of the time consumption of clustering & region growth as already shown in Fig. 6 in the manuscript. The two processes are not the main source of delay. For DBSCAN process, it takes 2.3 ms out of 65.1 ms (3.5%) in *KITTI* dataset, 1.3 ms out of 64.6 ms (2.0%) in *Waymo* dataset, 0.8 ms out of 17.6 ms (4.5%) in *nuScenes* dataset, and 0.2 ms out of 11.5 ms (1.7%) in *AVIA-Indoor* dataset. For RANSAC process, it takes

Figure R2: **The top view of LiDAR scanning.** From the top view, it’s clear that LiDAR scans the environment at a nearly uniform resolution in all directions, leading to the depth images used for occlusion check a 360° horizontal FoV and a nearly uniform point density regardless of the direction. Therefore, the size of the neighborhood (denoted by blue area) remains the same with a fixed threshold ε_h , no matter which angles the moving object is located in the LiDAR FoV.

3.1 ms out of 65.1 ms (4.8%) in *KITTI* dataset, 3.0 ms out of 64.6 ms (4.6%) in *Waymo* dataset, 0.1 ms out of 17.6 ms (0.6%) in *nuScenes* dataset, and 0.1 ms out of 11.5 ms (0.7%) in *AVIA-Indoor* dataset.

The reviewer is right that the original versions of DBSCAN and RANSAC have high computation costs. We optimized both two algorithms to make them efficient in M-detector as detailed below.

As introduced in the section of *Clustering and Region Growth*, clustering directly on event points could be computationally expensive when the point density is high. To address this issue, event points are voxelized (voxel size L_v). Those voxels containing event points are called event voxels. The number of event voxels is usually far fewer than the number of event points, leading to a consumption reduction for DBSCAN. What’s more, the voxels are uniformly cut which facilitates us to design a simplified DBSCAN algorithm. In the simplified DBSCAN process, the event voxels are iterated. If the current event voxel is connected with other event voxels, and the total number of connected voxels is greater than a threshold C_{min} , these connected voxels will be clustered and all the points contained in these voxels are seen as a clustered set. On the contrary, if the connection does not exceed C_{min} , the clustering fails and the event points contained in the event voxels will be re-labeled as non-event points. The time complexity of the original DBSCAN is $O(N \log N)$ while our implemented DBSCAN has a time complexity of $O(N)$. As mentioned above, the

number of event voxels is fewer than the number of event points, and the time complexity of clustering is decreased. Therefore, in our implementation, the DBSCAN did not cost too much computation resources.

The classical RANSAC algorithm randomly selects n points from total N candidate points as inliers to estimate a plane, leading the maximum estimating times to be C_N^n . To prevent the boosting computational consumption caused by repeatedly estimating planes, we divide the total N candidate points into N/n groups for plane estimation and the maximum estimation times reduce to N/n . Although this simplification makes the algorithm achieve a sub-optimal plane estimation result, it is sufficient to obtain a good ground plane estimation because the inlier points usually occupy a great portion of the total candidate points in our case. Other than this difference, the remaining process of our implemented RANSAC is identical to the classical RANSAC algorithm. As a result, the RANSAC algorithm used for ground estimation here is not time-consuming.

***Q4** Object tracking: The authors do not detect an object even if it is previously detected and pause its motion in a couple of seconds, which can be a major flaw in real-world applications such as collision avoidance in congested stop-and-go traffic or at traffic signals. Such flaws may be caused by their algorithm not including any tracking capability. If an object can be tracked, whether or not they become static will not be an issue. The authors may want to explore some complementary LiDAR tracking algorithms to address this issue.*

Reply: We appreciate the reviewer for the constructive comments. Yes, the reviewer is right that without a tracking algorithm, M-detector detects only objects that are currently moving, but not objects currently static but moved previously. We hope to clarify that such a feature is not a flaw of the algorithm, but the genuine aim of moving event detection (i.e., detecting points on objects that are moving at the current time, not objects that are movable nor objects that moved previously).

We also agree with the reviewer that in practical applications, moving objects with temporary stops (e.g., at the crosswalk) is a very common and typical scenario that should be addressed. One possible way is to combine M-detector with a tracking algorithm as suggested by the reviewer. Specifically, for any moving points predicted by M-detector at a frame, we initialize a tracker from that frame. Then in the subsequent frames, we use the raw points (not just event points) to update the tracker. Since the tracker can provide a fairly good location prediction, identifying points on the object from the raw points could be significantly accelerated. We have implemented a preliminary tracker based on this idea to provide proof of feasibility. One demonstration of the preliminary implementation on *KITTI* Sequence 11 is shown in a video at **Tracking Demonstration Video**.

Since the scope of this article is on moving event detection, we would like to focus on the

topic and discuss the potential tracking algorithm in Supplementary Section S1.2 (the first paragraph). A more thorough investigation on the tracking capability will be conducted in future work.

Q 5 *Model evaluation and calibration, Equation 1, Page 5: Since this method is a standard model-based method that involves the calibration of thresholds, the author should provide some threshold calibration results, such as the precision-recall curves or receiver operating characteristic curves. Some guidelines should be provided on how to calibrate those thresholds, such as neighborhood or occlusion thresholds, including its the feasible range, optimization methods, and the calibrated values for each sample application.*

Reply: We thank the reviewer for the constructive suggestion, following which, guidelines on how to calibrate key thresholds and the corresponding Precision-Recall curves have been provided.

In the guidelines, fourteen key parameters, as outlined in Table R1, including r_h , r_v , ε_h , ε_v , ε_d , ε_φ , ε_θ , ε_f , ε_b , N , M_1 , M_2 , M_3 , and L_v , are selected. These parameters’ potentially feasible ranges (as shown in Table R1) are provided and their impacts on M-detector’s performance are analyzed according to the M-detector’s design, as provided in Supplementary Section S8.

Table R1: Suggested ranges for parameters in M-detector.

Module	Symbol	Parameter explanation	Suggested range
Depth image construction	r_h	The horizontal pixel size (i.e., resolution) of the depth image	$a_h-2a_h^1$
	r_v	The vertical pixel size (i.e., resolution) of the depth image	$a_v-2a_v^2$
Occlusion check	ε_h	The neighborhood size in the horizontal direction	$0.5a_h-a_h$
	ε_v	The neighborhood size in the vertical direction	$0.5a_v-a_v$
	ε_d	The depth difference threshold	0.1-5.0 m
Map consistency check	ε_φ	The neighborhood size in azimuthal angle	$2a_h-8a_h$
	ε_θ	The neighborhood size in polar angle	$2a_v-8a_v$
	ε_f	The forward depth difference threshold	0.1-5.0 m
	ε_b	The backward depth difference threshold	0.1-5.0 m
Event tests	N	The number of latest depth images saved in depth image library	5-50
	M_1	The minimum number of depth images required in test one	1-5
	M_2	The minimum number of depth images required in test two	2-5
	M_3	The minimum number of depth images required in test three	2-5
Clustering	L_v	The voxel size	0.1-1.0 m

¹ a_h is the horizontal angular resolution of LiDAR.

² a_v is the vertical angular resolution of LiDAR.

Using sequence 10 in the *AVIA-Indoor* dataset as an example, we perturbed the value of one parameter in the range provided by Table R1 while keeping other parameters at their nominal values to explore its impact on the performance and plot the Precision-Recall curves. The results are illustrated in Fig. R3 and Fig. R4 with the nominal values set as the 1x base.

Figure R3: **The first group of Precision-Recall curves of M-detector on six parameters.** Results in blue are obtained when the M-detector runs in frame-out mode and results in green are obtained from the M-detector running in point-out mode. Each data point represents the total precision and recall over all frames in sequence 10 of *AVIA-Indoor*. The label of each point represents the multiple of the parameter’s current value relative to its nominal value (0.30° for r_h , 0.30° for r_v , 0.3 m for ε_d , 0.60° for ε_φ , 1.20° for ε_θ , and 1.0 m for ε_b).

Figure R4: **The second group of Precision-Recall curves of M-detector on four parameters.** Results in blue are obtained when the M-detector runs in frame-out mode and results in green are obtained from the M-detector running in point-out mode. Each data point represents the total precision and recall over all frames in sequence 10 of *AVIA-Indoor*. The label of each point represents the multiple of the parameter's current value relative to its nominal value (5 for N , 2 for M_1 , 3 for M_3 , and 0.3 m for L_v).

As shown in Fig. R3 and Fig. R4, M-detector can maintain consistently high performance over a large range of parameter values, which validates the robustness of our method against parameter values. Moreover, the performance trend for each parameter is consistent with the analysis as provided in Supplementary Section S8.1. More details on the calibration process and the corresponding results are provided in Supplementary Section S8. The specific content includes:

S8 Instructions for Parameters Tuning

M-detector is a model-based method, so its parameters must be tuned for good performance. To facilitate its use in the future, some guidance on parameter tuning is provided in this section. Fourteen parameters are selected to illustrate how to choose the appropriate values, as shown in Table S5, including the parameters for depth image construction (r_h and r_v), occlusion check (ε_h , ε_v , and ε_d), map consistency check (ε_φ , ε_θ , ε_f , and ε_b), clustering (L_v), and event tests (N , M_1 , M_2 , and M_3). The Precision-Recall curves are used for tuning and evaluation of each parameter. The computation for precision P and recall R is:

$$P = \frac{TP}{TP + FP}, R = \frac{TP}{TP + FN} \tag{1}$$

where TP, FP, FN denote the number of true positive, false positive, and false negative points on moving objects over all frames in the sequence used for tuning.

S8.1 Guidelines for parameters tuning

Parameters for depth image construction. Parameters for depth image construction include the horizontal pixel size (i.e., resolution) r_h and the vertical pixel size (i.e., resolution) r_v . In M-detector, as introduced in the section of *Occlusion check on depth image*, the occlusion check is achieved based on depth images. To determine if there is occlusion, each pixel should contain at least one LiDAR point. Therefore, the resolution of the depth image r_h and r_v is usually slightly larger than (e.g., 1-2 times) the LiDAR horizontal and vertical angular resolution, respectively, ensuring that there are points in each pixel. If r_h or r_v is too small, pixels may have no points due to the limited LiDAR angular resolution, making it impossible to determine the occlusion nor eventness of a point projected into this pixel. The missing detection of event points arising from this hence increases the FN. If the resolution is too large, a pixel will contain enough background points for occlusion determination, which helps the detection of event points and hence increases TP. Meanwhile, the large pixel size could also contain background points far from the test point, causing wrong event detection

that increases FP.

Parameters for occlusion check. Parameters for occlusion check include the neighborhood size in the horizontal direction ε_h , the neighborhood size in the vertical direction ε_v , and the depth difference threshold ε_d . For ε_h or ε_v , since occlusion check is based on depth image, the value is usually set to 0.5 times r_h and r_v . For ε_d , the value is usually within 0.1-5.0 m. If the value ε_d is too small, event points can pass the occlusion check more easily, resulting in a higher number of TP. But meanwhile, static points could also be tested positive for the occlusion check due to the LiDAR ranging errors (often a few centimeters), which leads to the increment of FP as well. Conversely, if the value is too large that requires the current point to have a large depth difference from the background, points on moving objects with relatively low moving speed cannot be detected, which results in the increment of FN.

Parameters for map consistency check. Parameters for map consistency check include the neighborhood size in azimuthal angle ε_φ , the neighborhood size in polar angle ε_θ , the forward depth difference threshold ε_f , and the backward depth difference threshold ε_b . For ε_φ or ε_θ , the value is usually set to 2-8 times the horizontal or vertical angular resolution of LiDAR. In M-detector, as outlined in the section of *map consistency check*, map consistency check is based on the principle that points very close to static map points should also be considered static (i.e., the point should be rejected if it is tested as an event). These two parameters define the neighborhood size (defined on depth images) of the map points that a candidate point should be examined against. If these two parameters are very small, only the very nearby static map points can reject an event point, resulting in more TP points. Meanwhile, the small neighborhood is less likely to reject wrong event points, which results in more FP points as well. If ε_φ or ε_θ is large, more static map points from an extended neighborhood can reject an event point, making it less likely for event points to pass the map consistency check successfully, resulting in more FN points.

For ε_f or ε_b , the value is usually within 0.1-5.0 m. If the value is smaller, a smaller distance between the event point and static points will be allowed. Less static points are used to reject the positive occlusion. Thus, more event points will pass the map consistency check, resulting in a higher number of TP points. Meanwhile, the static points mislabeled as events are also easier to pass map consistency check, contributing to a higher number of FP points. Conversely, when the value is larger, the minimum distance between the current point and the static points should be kept at a larger value, which makes map consistency check reject more ineffective occlusions, resulting in a higher number of FN points.

Parameters for event tests. Parameters for event tests include the number of latest depth images saved in depth image library N , the minimum number of depth images required in test one M_1 , the minimum number of depth images required in test two M_2 , and the

minimum number of depth images required in test three M_3 . The value of N is often between 5-50, depending on the affordable computation resources as larger N requires more computation. For M_1 , it must be greater than or equal to one while less than or equal to N . The suggested range for M_1 is 1-5. For M_2 or M_3 , it must be greater than or equal to two to ensure consecutive occlusions while less than or equal to N . The suggested range for M_2 or M_3 is 2-5.

Generally speaking, with a larger number of depth images N , more information in the past is utilized during the event detection, so the possibility of a test point passing the occlusion tests will be higher, leading to an increase in TP and FP. For the effect of M_1, M_2, M_3 , they are the minimum numbers of depth images required by the three tests, so a larger value will be more difficult for a test point to reach, resulting in a lower number of TP and FP points.

Parameter for clustering. The parameter for clustering includes the voxel size L_v in the voxelization process. This parameter is generally in the range of 0.1-1.0 m. As explained in the section of *Clustering and Region Growth*, clustering is performed on voxels containing event points (i.e., event voxels), then event points in isolated event voxels will be re-labeled as non-event points, and non-event points contained in voxel clusters of a certain number will also be re-labeled as event points. Reducing the voxel size will reduce the range for recalling non-event points, resulting in a decrease in TP. Conversely, with a larger voxel size, rejecting one isolated event pixel could reject more wrong event points (i.e., decrease in FP), but meanwhile reject more true event points contained in this voxel (i.e., increase in FN).

S8.2 Verification and results

According to the analysis presented in the previous section, the parameters' values can be tuned and the results for datasets used in the benchmark are shown in Table S2. Factually, we found that M-detector's performance is quite robust to the parameter values, requiring no precise tuning in practice. Taking the *AVIA-Indoor* dataset sequence 10 for example, we plot the Precision-Recall curves to investigate the performance change versus the parameter values. For each parameter under investigation, we perturb its values from its nominal values (i.e., the last row of Table S2) while keeping all other parameters at their respective nominal values to obtain a Precision-Recall curve.

The Precision-Recall curves are presented in Fig. S7 and Fig. S8. It's noted that given that $\varepsilon_h/\varepsilon_v$ is usually set to 0.5 times the depth image resolution r_h/r_v , ε_b is usually set to ε_d , and the tuning methods of M_2 and M_3 are identical, so we only investigate the Precision-Recall for 10 independent parameters, including $r_h, r_v, \varepsilon_d, \varepsilon_\varphi, \varepsilon_\theta, \varepsilon_f, N, M_1, M_3$, and L_v .

For parameters for depth image construction r_h and r_v , as illustrated in Fig. S7, the

1x denoted the nominal value 0.30° and 0.30° respectively. The recall increased as the value became larger, indicating more TP points (note that the sum of TP points and FN points is fixed for a specific sequence), which agrees with the analysis in the previous section. Meanwhile, the precision slightly increased (in point-out mode) with the larger r_h and r_v too, which is caused by the larger increase in TP than in FP, a trend consistent with the previous analysis as well. Moreover, M-detector achieved relatively high performance (precision above 0.86 and recall above 0.7) in a large range of parameter values (0.5x-12.0x for r_h and 0.5x-6.0x for r_v).

For parameters for occlusion check ε_d , as illustrated in Fig. S7, the 1x denoted the nominal value 0.3 m. With its value smaller, the recall increased, which means there are more TP points and fewer FN points, matched with the analysis in the previous section. The precision slightly increased because the increase rate of TP is slightly larger than FP's. Besides, M-detector maintained relatively high performance (precision above 0.86 and recall above 0.75) in a large range of parameter values (0.33x-16.67x for ε_d).

For parameters for map consistency check ε_φ , ε_θ , and ε_f , as illustrated in Fig. S7, the 1x denoted the nominal value 0.6° , 1.2° , and 1.0 m, respectively. The recall increased and precision decreased with the value smaller, indicating more TP points, fewer FN points, and more FP points, which is consistent with the analysis in the previous section. Besides, M-detector maintained relatively high performance (precision above 0.75 and recall above 0.70) in a large range of parameters values (0.5x-10.0x for ε_φ , 0.25x-4.0x for ε_θ , and 0.1x-5.0x for ε_f).

For parameters for event tests N , M_1 , and M_3 , as illustrated in Fig. S8, the 1x denoted the nominal value 5, 2, and 3, respectively. The recall increased and precision decreased as the value N was larger while M_1 and M_3 were smaller, indicating more TP points and fewer FN points, which verified the analysis in the previous section. Besides, M-detector maintained relatively high performance (precision above 0.82 and recall above 0.70) in a large range of parameter values (1x-10x for N , 0.5x-1.5x for M_1 , and 0.67x-1.67x for M_3).

For the parameter for clustering L_v , as illustrated in Fig. S8, the 1x base denoted the nominal value 0.30 m. As the value became larger, the recall initially increased and then decreased, which means TP increased first and then decreased while FN is verse, which agrees with the analysis in the previous section. For precision, it increased with the L_v at first, which means at those values, the clustering brought back more TP points than FP, corresponding with the analysis in the previous section for L_v . It was necessary to choose a suitable value for voxelization to ensure high recall while maintaining high precision. Also, M-detector maintained relatively high performance (precision above 0.88 and recall above 0.82) in a large range of the parameter value (0.17x and 3.33x for L_v).

In conclusion, the Precision-Recall curves in Fig. S7 and Fig. S8 demonstrated that M-

detector can maintain relatively high performance over a relatively large range of parameter values, and the tuning curves for each parameter are consistent with the above analysis in Section S8.1.

Q 6 *Other minor comments: Line 523 and 537, Page 14: “depth” is never clearly defined through equations 2-6. It kept being referred to without further details. How was depth calculated for a point? Which coordinate system is it in? Is it a vector with three elements? Is d in Equation 5 the depth?*

Reply: Yes, d in Equation 5 represents the depth, which is the radial distance of a point in the depth image reference frame. It is a scalar, not a vector. We apologize for the missing definition for this term and have revised the manuscript to clarify that the radial distance calculated by Equation 5 is depth. The specific content is:

“ φ , θ , and d represent the point’s azimuthal angle, polar angle, and radial distance (i.e., depth) in spherical coordinates, respectively.”

Q 7 *Lines 257-260, Page 7: Please provide the processing time of the M-detector also in microseconds, not just relative percentages of the computational time of LMNet and other comparison algorithms.*

Reply: Regarding the processing time of M-detector, we have actually provided the processing time of M-detector in microseconds in the section of *Time consumption and detection latency* of the original manuscript already. Specifically, the processing time for the *KITTI* dataset is 87.3 ms, for the *Waymo* dataset is 86.2 ms, for the *nuScenes* dataset is 33.6 ms, and for the *AVIA-Indoor* dataset is 16.1 ms. These values are listed together with the frame period to demonstrate the real-time capability of M-detector. The content in the manuscript is : “Overall, M-detector consumed only around 1-6% computation time of LMNet-1 when running on the same CPU, and only around 20-60% of LMNet-1 even when LMNet-1 additionally used a GPU. The resultant time consumption of the M-detector was less than the frame period (87.3 ms versus 100 ms in *KITTI*, 86.2 ms versus 100 ms in *Waymo*, 33.6 ms versus 50 ms in *nuScenes*, and 16.1 ms versus 100 ms in *AVIA-Indoor*), suggesting a real-time running performance.”

References

- [1] R. Massaro, J. Anderson, J. Nelson, and J. Edwards, “A comparative study between frequency-modulated continuous wave lidar and linear mode lidar.” *International Archives of the Photogrammetry, Remote Sensing & Spatial Information Sciences*, 2014.

- [2] J. Anderson, R. Massaro, J. Curry, R. Reibel, J. Nelson, and J. Edwards, “Ladar: frequency-modulated, continuous wave laser detection and ranging,” *Photogrammetric Engineering & Remote Sensing*, vol. 83, no. 11, pp. 721–727, 2017.

REVIEWER COMMENTS

Reviewer #1 (Remarks to the Author):

The author has carefully answered the comments of the reviewers, and I suggest accepting and publishing this paper.

Reviewer #2 (Remarks to the Author):

Thank you for your insights and answers to my previous questions and concerns.

The value of the occlusion approach was made evident as a possible competitor to physical implementations of moving event detection such as FMCW Lidar and some of the smaller clarifications were satisfactorily addressed.

I support that this paper should be considered for publication at the discretion of the editor.

Reviewer #3 (Remarks to the Author):

The authors made significant efforts to address the comments. Most of the comments have been addressed. Here are some remaining minor comments:

Q2-Response. The authors may want to do some further tests in open space. Often, at the boundaries of the LiDAR sensor range, the reflection points are not so stable, even with solid surfaces. It becomes even worse if there is vegetation. The variance of the depth measurements will vary significantly between those boundary/vegetation locations and nearby ground or infrastructure locations. A uniform threshold for different depths or reflection surface conditions will not be robust. However, the above issues will be alleviated if these thresholds are individually trained and calibrated for each ϕ and θ angle combinations in the space.

Q3-Response: Thanks for the explanation on the improved RANSAC and DBSCAN algorithms with discretization. Meanwhile, given that RANSAC and DBSCAN are not the main delay sources for this algorithm, can the authors provide a breakdown of what some of the critical module's computational time and cost are, especially those modules critical to the remaining 65ms? Most of the consumption is on browsing through all the points? The proposed algorithm is much faster than many existing algorithms, but it will be helpful to provide a breakdown of those time-consuming modules.

Other comments have been addressed adequately. Thanks for the effort.

Response to Reviewer Comments

September 14, 2023

We greatly appreciate all the helpful comments, and have done our best to carefully address them. Below, [Q] is used to indicate questions or comments from the reviewer, and [Reply:] is used to indicate our corresponding answers. Please note that cited texts from our manuscript are presented in blue respectively. The reviewer's comments have been listed in order below in the italic text, followed by our responses in normal text.

REVIEWER COMMENTS

Reviewer #1 (Remarks to the Author):

The author has carefully answered the comments of the reviewers, and I suggest accepting and publishing this paper.

Reviewer #2 (Remarks to the Author):

Thank you for your insights and answers to my previous questions and concerns. The value of the occlusion approach was made evident as a possible competitor to physical implementations of moving event detection such as FMCW Lidar and some of the smaller clarifications were satisfactorily addressed.

I support that this paper should be considered for publication at the discretion of the editor.

Reviewer #3 (Remarks to the Author):

The authors made significant efforts to address the comments. Most of the comments have been addressed. Here are some remaining minor comments:

Q2-Response. The authors may want to do some further tests in open space. Often, at the boundaries of the LiDAR sensor range, the reflection points are not so stable, even with solid surfaces. It becomes even worse if there is vegetation. The variance of the

depth measurements will vary significantly between those boundary/vegetation locations and nearby ground or infrastructure locations. A uniform threshold for different depths or reflection surface conditions will not be robust. However, the above issues will be alleviated if these thresholds are individually trained and calibrated for each ϕ and θ angle combinations in the space.

Q3-Response: Thanks for the explanation on the improved RANSAC and DBSCAN algorithms with discretization. Meanwhile, given that RANSAC and DBSCAN are not the main delay sources for this algorithm, can the authors provide a breakdown of what some of the critical module's computational time and cost are, especially those modules critical to the remaining 65ms? Most of the consumption is on browsing through all the points? The proposed algorithm is much faster than many existing algorithms, but it will be helpful to provide a breakdown of those time-consuming modules.

Point-by-point Responses to the Reviewers' Comments

Reviewer 1

Q 1 *The author has carefully answered the comments of the reviewers, and I suggest accepting and publishing this paper.*

Reply: We appreciate the reviewer's reviewing efforts and the constructive feedback.

Reviewer 2

Q1 Thank you for your insights and answers to my previous questions and concerns. The value of the occlusion approach was made evident as a possible competitor to physical implementations of moving event detection such as FMCW Lidar and some of the smaller clarifications were satisfactorily addressed. I support that this paper should be considered for publication at the discretion of the editor.

Reply: We appreciate the reviewer's reviewing efforts and the constructive feedback.

Reviewer 3

Q1 The authors made significant efforts to address the comments. Most of the comments have been addressed. Here are some remaining minor comments:

Reply: We are grateful to the reviewer’s reviewing efforts and recognition of our work. We carefully considered the reviewer’s comments and made the necessary revisions to further improve our work. We hope this response satisfactorily addresses the reviewer’s comments.

Q2 Q2-Response. The authors may want to do some further tests in open space. Often, at the boundaries of the LiDAR sensor range, the reflection points are not so stable, even with solid surfaces. It becomes even worse if there is vegetation. The variance of the depth measurements will vary significantly between those boundary/vegetation locations and nearby ground or infrastructure locations. A uniform threshold for different depths or reflection surface conditions will not be robust. However, the above issues will be alleviated if these thresholds are individually trained and calibrated for each ϕ and θ angle combinations in the space.

Reply: We thank the reviewer for raising this point. The reviewer’s concerns are reasonable. As the depth increases, various objects at the boundary of the LiDAR can result in inconsistent depth measurements. One simple and straightforward solution to this problem is to truncate the points that are too far (with depth beyond a certain value). Such truncation does not harm the overall system performance in view of the long sensing range of a LiDAR sensor, where moving objects at far does not require timely detection and reaction of the ego-vehicle.

If it is truly desired to utilize the points at the boundary, as mentioned by the reviewer, calibrating different thresholds for each combination of ϕ and θ angles in the LiDAR’s field of view might be effective in mitigating this issue and improving the performance of the M-detector. Nevertheless, this approach presents an inherent challenge because the location (i.e., ϕ and θ angles) of objects (e.g., sky, vegetation, ground) causing large depth variation are not known in advance. As a consequence, calibrating the thresholds for different ϕ and θ angles is not possible. Even though this can be done for a certain scenario (e.g., self-driving) where the locations of grounds, vegetation, and sky in the sensor FoV are roughly fixed, it is not very accurate and could make the performance rather sensitive to individual scenes. The overfitted parameters are also hard to be generalized to other scenarios.

Instead, a more suitable approach is to set different thresholds for different depth, considering that the depth measurements variation is caused by distance. Following the reviewer’s suggestion, we implemented adaptive thresholds for depth comparison used in occlusion check

and map consistency check based on different points distance. The greater the depth measurement variation is, the larger the depth comparison threshold should be to reduce the impact of such variation. Thus, in our implementation, we deployed a threshold that is linearly increasing with respect to the points distance, i.e., $\min(d_{max}, d_i + \max(0, k_{thr} * (d - d_{thr})))$, where d_{max} is the maximum value of the threshold, d_i represents the initial depth threshold used in occlusion check (i.e., ε_d) or map consistency check (i.e., ε_f and ε_b), k_{thr} denotes the linear increasing rate, d_{thr} represents the distance the threshold starts increasing, and d is the distance of the tested point. When the point distance is below d_{thr} , a uniform threshold remains used. When the distance exceeds d_{thr} , the threshold increases linearly in proportion to the distance increment, with the increase rate k_{thr} , until reaching the maximum value d_{max} . To verify the effectiveness of this method, we performed tests on the *KITTI* dataset because its sequences 00-07 contain highway scenes in which most of the data were collected in open space and LiDAR scanned objects located at a relatively further distance. To determine suitable values for k_{thr} , d_{thr} , and d_{max} , several tests were conducted and the values of 0.1, 50, and, 6 were selected, respectively.

The detailed results comparing the performance of the adaptive threshold and uniform threshold in both point-out and frame-out modes are presented in Table R1 and R2. Overall, the adaptive threshold demonstrated better performance than the uniform threshold in point-out mode. Table R1 clearly demonstrates a consistent improvement in the Intersection over Union (IoU). Out of the 20 sequences tested, the IoU increased in 17 of them and also the total IoU. However, in frame-out mode shown in Table R2, the adaptive threshold either had no significant effect (in 8 out of 20 sequences) or even had a negative impact (in 10 out of 20 sequences), leading to an overall IoU comparable to that of the uniform threshold.

The above results can be explained as follows. Due to the linear increment from the initial threshold (the value taken by the uniform threshold strategy), the adaptive threshold has a larger threshold value, making it more difficult to be considered as positive (less FP and TP). In point-out mode, the eventness of each raw point is determined immediately after arrival while leveraging no extra information (e.g., point density), resulting in a certain number of FP points. Therefore, reducing such number of FP using adaptive thresholds could effectively bring up IoU (see Fig. R1a and R1b). In contrast, in frame-out mode, a large number of FP points have already been eliminated by the step of clustering & region growth, so the benefits brought by adaptive threshold are not evident. Instead, the reduced number of TP caused by adaptive threshold could lower the overall IoU (see Fig. R1c and R1d).

Based on the results and analysis provided, it is obvious that the adaptive threshold generally proves to be beneficial in point-out mode. However, in frame-out mode, the adaptive threshold either has no significant effect or even exhibits a negative impact on performance.

Therefore, the specific choice of threshold depends on the specific requirements and objectives of the application. The above analysis has been added in Supplementary Section S8.3.

Table R1: **Comparison of adaptive threshold and uniform threshold on *KITTI* dataset in point-out mode.**

Sequences	Adaptive threshold				Uniform threshold			
	TP	FP	FN	IoU	TP	FP	FN	IoU
00	99,821	17,201	26,498	0.696	99,899	19,732	26,420	0.684
01	136,282	26,051	59,973	0.613	137,187	35,813	59,068	0.591
02	72,229	27,022	5,257	0.691	72,482	36,487	5,004	0.636
03	147,442	73,814	47,890	0.548	148,260	100688	47072	0.501
04	199,646	48,226	48,536	0.674	200,421	80,698	47,761	0.609
05	309,884	22,002	74,161	0.763	313,212	33,799	70,833	0.750
06	139,312	66,684	59,043	0.526	139,698	90,662	58,657	0.483
07	140,823	60,388	55,587	0.548	152,029	114,496	44,381	0.489
08	171,939	122,063	118,753	0.417	187,793	183,773	102,899	0.396
09	166,088	40,136	301,878	0.327	166,414	65,242	301,552	0.312
10	239,200	62,835	39,242	0.701	240,211	84,772	38,231	0.661
11	10,099	666	3,424	0.712	10465	1,109	3,058	0.715
12	121,929	37,475	29,691	0.645	122,083	51,410	29,537	0.601
13	19,450	13,280	3,499	0.537	19,924	16,863	3,025	0.500
14	150,268	21,203	14,878	0.806	151,209	29,902	13,937	0.775
15	320,406	12,772	46,148	0.845	320,419	19,149	46,135	0.831
16	248,111	28,491	22,184	0.830	248,111	28,738	22,184	0.830
17	461,347	61,131	115,235	0.723	462,426	81,298	114,156	0.703
18	1,375,009	162,899	473,763	0.684	1,375,836	180,200	472,936	0.678
19	997,415	126,472	1,313,366	0.409	1,022,951	162,074	1,287,830	0.414
Total	5,526,700	1,030,811	2,859,006	0.587	5,591,030	1,416,905	2,794,676	0.570

Table R2: Comparison of adaptive threshold and uniform threshold on *KITTI* dataset in frame-out mode.

Sequences	Adaptive threshold				Uniform threshold			
	TP	FP	FN	IoU	TP	FP	FN	IoU
00	106,383	2,461	19,936	0.826	106,407	2,552	19,912	0.826
01	147,147	6,934	49,108	0.724	147,703	7,241	48,552	0.726
02	72,687	8,439	4,799	0.846	72,799	8,521	4,687	0.846
03	153,875	8,345	41,457	0.755	154,145	8,457	41,187	0.756
04	233,339	4,955	14,843	0.922	233,836	5,001	14,346	0.924
05	358,581	5,080	25,464	0.922	361,241	5648	22,804	0.927
06	168,067	13,032	30,288	0.795	168,204	13,427	30,151	0.794
07	139,504	2,728	56,906	0.700	149,958	4,401	46,452	0.747
08	188,688	9,706	102,004	0.628	198,740	12,072	91,952	0.656
09	188,146	5,049	279,820	0.400	187,225	5,659	280,741	0.400
10	257,819	14,488	20,623	0.880	258,236	14,952	20,206	0.880
11	10,121	635	3,402	0.715	10,222	647	3,301	0.721
12	129,192	14,736	22,428	0.777	129,301	15,730	22,319	0.773
13	1,8516	1,551	4,433	0.756	18,718	1,720	4,231	0.759
14	147,231	11,395	17,915	0.834	147,770	11,795	17,376	0.835
15	329,054	10,254	37,500	0.873	329,195	10,665	37,359	0.873
16	257,796	21,690	12,499	0.883	257,796	21,690	12,499	0.883
17	551,655	26,947	24,927	0.914	551,884	27,401	24,698	0.914
18	1,468,764	143,906	380,008	0.737	1,469,605	144,684	379,167	0.737
19	1,384,019	50,614	926,762	0.586	1,403,820	55,454	906,961	0.593
Total	6,310,584	362,945	2,075,122	0.721	6,356,805	377,717	2,028,901	0.725

Figure R1: **The comparison results between adaptive threshold and uniform threshold.** The point cloud is from the 207th frame in sequence 05 of *KITTI*. The red, green, and purple points denote true positive (TP), false positive (FP), and false negative (FN) points respectively. Boxes in detection results are manually labeled to highlight the regions of interest. **a** The detection result of adaptive threshold in point-out mode. **b** The detection result of uniform threshold in point-out mode. **c** The detection result of adaptive threshold in frame-out mode. **d** The detection result of uniform threshold in frame-out mode.

Q3 *Q3-Response: Thanks for the explanation on the improved RANSAC and DBSCAN algorithms with discretization. Meanwhile, given that RANSAC and DBSCAN are not the main delay sources for this algorithm, can the authors provide a breakdown of what some of the critical module’s computational time and cost are, especially those modules critical to the remaining 65ms? Most of the consumption is on browsing through all the points? The proposed algorithm is much faster than many existing algorithms, but it will be helpful to provide a breakdown of those time-consuming modules.*

Reply:

Thanks for the reviewer’s suggestion. As depicted in Figure 6b of the original manuscript, the detection latency consists of two steps: event detection and clustering & region growth. Both these two modules need to browse each individual raw point, which indeed contributes to the most time consumption as mentioned by the reviewer, due to the large number of raw points to browse. In event detection, three sub-steps are conducted as presented in Figure 4b of the original manuscript: projecting the points into the previous depth images, occlusion check, and map consistency check. In clustering & region growth, two sub-steps are performed as depicted in Section Methods of the original manuscript: clustering, and region growth. In response to the reviewer’s suggestion, we conducted further analysis of the time consumption proportion of these five sub-steps versus the total detection latency. The results are shown in Table R3.

As can be seen, among the five sub-steps, the projection and map consistency check steps consume the most time. Specifically, for projection, it takes 39.38% of the detection latency in *KITTI* dataset, 50.79% in *Waymo* dataset, 68.07% in *nuScenes* dataset, and 29.69% in *AVIA-Indoor* dataset. For map consistency check, it takes 38.21% in *KITTI* dataset, 36.91% in *Waymo* dataset, 19.01% in *nuScenes* dataset, and 52.70% in *AVIA-Indoor* dataset.

Following the reviewer’s suggestion, we have added the results of time consumption breakdown for event detection and clustering & region growth to Supplementary Table S4, which is referred in Section Results of the revised manuscript, cited as below:

“As shown in Fig. 6b, in the frame-out mode, M-detector had a detection latency ranging from 11.5 ms to 65.1 ms due to the varying frame rate and point number in each frame (see further latency breakdown in Supplementary Table S4).”

Table R3: **Computation time breakdown of event detection and clustering & region growth in M-detector on different datasets.** In the table, the number represents the percentage of the time consumption of this part over the total detection latency (consisting of both event detection and clustering & region growth).

	Event detection			Clustering & region growth	
	Projection	Occlusion check	Map consistency check	Clustering	Region growth
KITTI	39.38%	20.17%	38.21%	1.99%	0.24%
Waymo	50.79%	10.03%	36.91%	1.92%	0.35%
nuScenes	68.07%	9.53%	19.01%	3.01%	0.38%
AVIA-Indoor	29.69%	15.64%	52.70%	1.92%	0.05%

REVIEWERS' COMMENTS

Reviewer #3 (Remarks to the Author):

The authors have addressed my concerns. I am good with the responses and revision.

Response to Reviewer Comments

October 6, 2023

We greatly appreciate all the helpful comments, and have done our best to carefully address them. Below, [Q] is used to indicate questions or comments from the reviewer, and [Reply:] is used to indicate our corresponding answers. Please note that cited texts from our manuscript are presented in blue respectively. The reviewer's comments have been listed in order below in the italic text, followed by our responses in normal text.

REVIEWER COMMENTS

Reviewer #3 (Remarks to the Author):

The authors have addressed my concerns. I am good with the responses and revision.

Point-by-point Responses to the Reviewers' Comments

Reviewer 1

Q1 The authors have addressed my concerns. I am good with the responses and revision.

Reply: We appreciate the reviewer's reviewing efforts and the constructive feedback.